# HR repair pathway plays a crucial role in maintaining neural stem cell fate under irradiation stress

Xiao Xu[1], Huanping An[1,2] , Cheng Wu[1], Rong Sang[1], Litao Wu[1], Yuhan Lou[1], Xiaohang Yang[1,3] , Yongmei Xi[1]

Environmental stress can cause mutation or genomic instability in stem cells which, in some cases, leads to tumorigenesis. Mechanisms to monitor and eliminate these mutant stem cells remain elusive. Here, using the *Drosophila* larval brain as a model, we show that X-ray irradiation (IR) at the early larval stage leads to accumulation of nuclear Prospero (Pros), resulting in premature differentiation of neural stem cells (neuroblasts, NBs). Through NB-specific RNAi screenings, we determined that it is the Mre11–Rad50–Nbs1 complex and the homologous recombination (HR) repair pathway, rather than non-homologous end-joining pathway that plays, a dominant role in the maintenance of NBs under IR stress. The DNA damage sensor ATR/*mei-41* is shown to act to prevent IR-induced nuclear Pros in a WRNexo-dependent manner. The accumulation of nuclear Pros in NBs under IR stress, leads to NB cell fate termination, rather than resulting in mutant cell proliferation. Our study reveals an emerging mechanism for the HR repair pathway in maintaining neural stem cell fate under irradiation stress.

## Introduction

DNA damage caused by ionizing radiation (IR) can induce cell cycle arrest, senescence, differentiation, or cell death, all of which affect tissue homeostasis. It has been estimated that somewhere between a 1,000 to a million DNA damage repair events occur every 24 h per cell, either related to general metabolism or as response to external stresses such as radiation (Lindahl & Barnes, 2000; Carusillo & Mussolino, 2020). Such DNA damage repair events are specifically noted to occur during early development in various types of stem cells (Vitale et al, 2017). Among the numerous DNA damage repair events in living cells, a small number of errors that occur through accumulation can result in genetic mutations. Genomic instability, as caused by DNA damage repair dysfunction, often changes the processes of proliferation, differentiation, and/ or dedifferentiation in stem cells, leading to altered stem cell fate,

termination, or tumorigenesis (Jackson & Bartek, 2009). The overall regulatory process of spontaneous or IR induced or DNA damage repair in neural stem cells remains unclear.

The *Drosophila* neural stem cell, also termed neuroblast (NB), is an ideal model to study stem cell characteristics (Doe, 2017; Wu et al, 2019; Sood et al, 2021). NBs are generated at the embryonic stage and terminated at the early pupal stage (Maurange et al, 2008; Walsh & Doe, 2017; Alvarez & Diaz-Benjumea, 2018). From birth to termination, NBs undergo multiple rounds of asymmetric division to produce sufficient daughter cells to form neural circuitries (Homem et al, 2015; Kang & Reichert, 2015). Accordingly, distinct NBs can be identified as either embryonic NBs, larval NBs, or pupal NBs (Truman & Bate, 1988; Ito & Hotta, 1992). Among these, larval NBs are composed of two groups, namely, central brain NBs and ventral nerve cord NBs (Boone & Doe, 2008). In the central brain, there are about 100 NBs per hemisphere lobe. During the development of the *Drosophila* central nervous system, tremendous stress can present fatal risk, generating considerable DNA damage to the NBs of larval brains. When resulting mutations occur, and neurogenesis is disturbed, NBs will often undergo premature termination (Doe, 2017; Hakes & Brand, 2019; Sood et al, 2021).

Pros, a homeodomain transcriptional factor expressed in NBs, primarily locates to the cytoplasm (Li & Vaessin, 2000; Chia et al, 2001). However, after each round of the NB asymmetric divisions, Pros is segregated into smaller daughter cells called ganglion mother cells (GMCs). Pros then enters the nucleus to promote GMC differentiation (Chia et al, 2001; Betschinger & Knoblich, 2004; Liu et al, 2020). In terminating NBs, Pros will enter the nucleus of NBs and initiate a final symmetrical division, producing two daughter cells (neurons or glia) that themselves do not subsequently divide (Maurange et al, 2008; Chai et al, 2013; Wu et al, 2019). Thus, terminating NBs can be clearly labeled by the presence of nuclear Pros (Chia et al, 2001; Chai et al, 2013; Wu et al, 2019; Sood et al, 2021). When stem cells experience external stress or genomic damage, Pros also skips its inherited temporal expression pattern, undergoing early differentiation and terminating the fate of NBs (Wagle & Song, 2020; Sang et al, 2022). This suggests that in the intrinsic mechanism by which neural

[1]The Women's Hospital, Institute of Genetics, Zhejiang Provincial Key Laboratory of Genetic & Development Disorders, School of Medicine, Zhejiang University, Hangzhou, China   [2]Key Laboratory of Clinical Molecular Biology of Hanzhong City, Hanzhong Vocational and Technique College, Hanzhong, China   [3]Joint Institute of Genetics and Genomic Medicine, Between Zhejiang University and University of Toronto, Zhejiang University, Hangzhou, China

Correspondence: xyyongm@zju.edu.cn; xhyang@zju.edu.cn

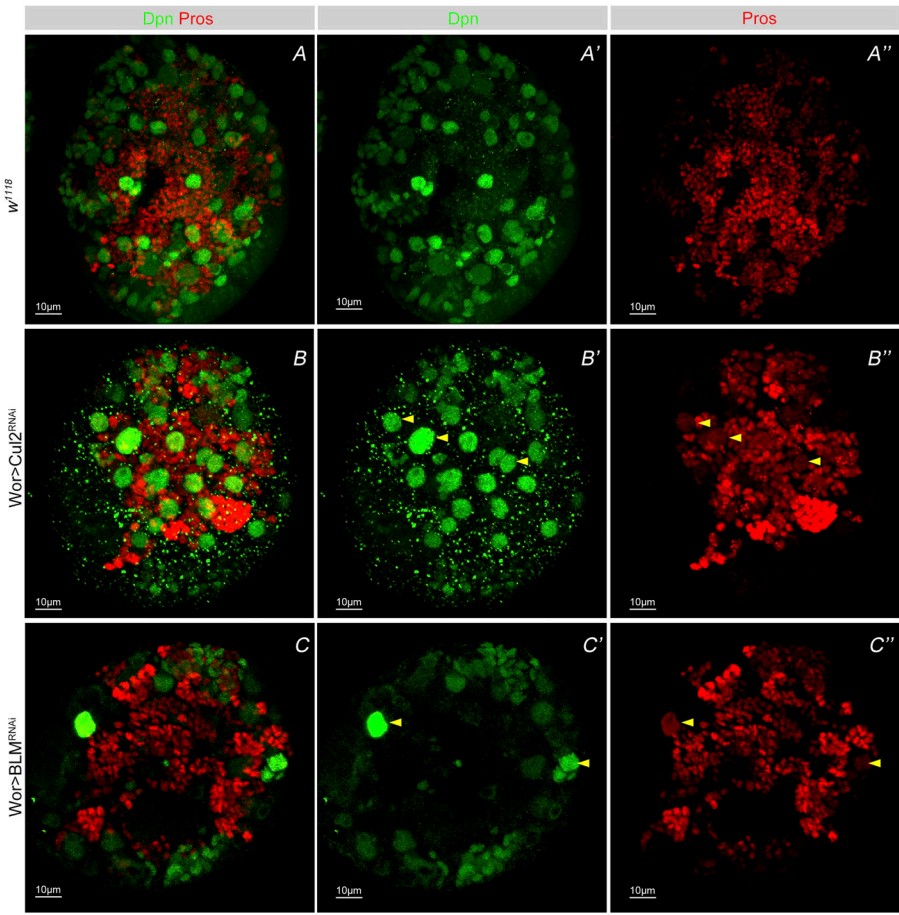

**Figure 1. Nuclear Pros as a marker for NB early differentiation and regulated by DNA damage repair genes.**
**(A–A")** Immunostaining of neuroblasts of wild-type third instar larvae and staining of third instar larval brains with the neuroblast marker Deadpan (Dpn) (green) and transcriptional factor Prospero (Pros) (A–A"). **(B–B", C–C")** Knocking down candidate genes from screening in third larval CNS (B–B", C–C"), resulted in premature Pros accumulation in the nucleus (yellow arrows). 5–10 larval brains of each genotype were dissected and observed.

stem cells terminate their NB cell fate, nuclear Pros plays a fundamental role.

During cell proliferation, abnormal nucleosome assembly can lead to chromatin breakage and chromatin structural instability. The nucleosome is composed of four core proteins (H2A, H2B, H3 and H4), which package about 147 bp of DNA fragments (Clapier et al, 2008). We previously found that the loss of function of the histone locus body protein Mxc led to premature accumulation of nuclear Pros and the termination of NBs stemness (Sang et al, 2022). In vitro studies have also observed that stem cells can enter ageing (senescence) or differentiate to avoid damage. Similarly, dysplastic stem cells are noted to transition into cancer cells when the chromosome is affected by ionizing radiation such as X-ray or chemical mutagen factors beyond a certain DNA damage repair threshold (Barazzuol et al, 2017).

The DNA damage repair process consists of two major steps, namely, DNA damage sensing, and DNA damage repair (Jackson & Bartek, 2009). Upon irradiation, double-stranded breaks (DSBs) occur, ataxia–telangiectasia mutated (ATM), ataxia–telangiectasia, and Rad3 related (ATR), then sense and respond to this DNA damage (Blackford & Jackson, 2017). Activated ATM and ATR can phosphorylate various substrates including the DNA damage indicator H2A histone family member X, and activate checkpoint kinase 1 (CHK1) and checkpoint kinase 2 (CHK2) (Awasthi et al, 2015). ATM–CHK2 and ATR–CHK1 stimulate the activity of the tumor suppressor genep53, which

promotes cell cycle arrest for DNA repair or regulated cell death if the DNA damage is beyond full recovery (LaRocque et al, 2007; Joyce et al, 2011; Chen, 2016). DSBs repair then follows as a second stage. Typical DSB repair pathways are composed of the non-homologous end-joining (NHEJ) pathway and homologous recombination (HR) repair pathway. In the DNA repair process, NHEJ and HR repair pathways are complementary, where the HR pathway is able to compensate when NHEJ is nonfunctional. In this case, the MRN (Mre11–Rad50–Nbs1) complex resects the DNA ends and initiates the HR repair pathway (Polo & Jackson, 2011; Reginato & Cejka, 2020).

In the present study, we first performed NB-specific RNAi screening and found that DNA damage repair genes are involved in the maintenance of *Drosophila* larval brain NBs. By exposing the *Drosophila* larvae to IR, we performed a second set of RNAi screening and demonstrated that the disturbance of DNA repair pathways by RNAi knock down, or by dysfunction combined with X-ray irradiation, leads to premature differentiation of NBs. The MRN complex and the HR repair pathways are then noted to play a crucial role in regulating the nuclear Pros in NBs under IR stress. The DNA damage sensor ATR/mei-41 is also clearly involved in this process in a Werner exonuclease (WRNexo)–dependent manner. Our study provides new insight into the understanding of the mechanism involved in neural stem cell maintenance under environmental stress.

**Table 1.   RNAi screening for genes with nuclear Pros Phenotype in NBs.**

| Gene | CG NO | Functions | Related process |
|------|-------|-----------|-----------------|
| Nup358 | CG11856 | Protein import into nucleus | Nucleocytoplasmic transport |
| smt3 | CG4494 | The only Drosophilla SUMO family protein, protein import into nucleus | Nucleocytoplasmic transport |
| Nup98-96 | CG10198 | Nucleocytoplasmic transporter activity | Nucleocytoplasmic transport |
| Nup160 | CG4738 | Nucleocytoplasmic transporter activity | Nucleocytoplasmic transport |
| Nup107 | CG6743 | Nucleocytoplasmic transporter activity | Nucleocytoplasmic transport |
| Mbo | CG6819 | Transporter activity | Nucleocytoplasmic transport |
| AP-1 M | CG9388 | Protein transporter activity | Nucleocytoplasmic transport |
| BLM | CG6920 | Repair replication fork damage and double strand | DNA damage repair |
| CG4078 | CG4078 | DNA repair | DNA damage repair |
| mre11 | CG16928 | DNA double-strand break repair protein | DNA damage repair |
| crm | CG2714 | Chromatin binding, regulation of transcription | Transcription |
| not1 | CG34407 | CCR4-NOT transcription complex subunit 1 | Transcription |
| mbd-R2 | CG10042 | DNA binding; protein binding | Transcription |
| Dref | CG5838 | Transcription factor activity | Transcription |
| SSRP | CG4817 | Regulation of DNA binding | Transcription |
| CG7386 | CG7386 | Regulation of transcription | Transcription |
| mute | CG34415 | Regulation of transcription | Transcription |
| lwr | CG3018 | Encodes Ubc9, a SUMO-conjugating enzyme | Protein catabolic process |
| Rca1 | CG10800 | Ubiquitin-protein transferase activity | Protein catabolic process |
| Cul-2 | CG1512 | Ubiquitin-protein ligase activity | Protein catabolic process |
| APC10 | CG11419 | Ubiquitin-protein ligase activity | Protein catabolic process |
| Ron12 | CG4157 | Ubiquitin-dependent protein catabolic process | Protein catabolic process |
| Ecd | CG5714 | Regulation of metabolic process | Metabolic process |
| AsnRS | CG10687 | Asparaginyl-tRNA aminoacylation | |
| cdc2 | CG5363 | Protein kinase activity, regulation of cell cycle process | Cell cycle |
| mars | CG10764 | Regulation of cell cycle | Cell cycle |

The screening result shows several of candidate genes, the suppression of which could cause Pros to become accumulated in the nucleus of NBs in the larval brain. The candidate genes could be separated into three main categories that relate to nucleocytoplasmic transportation, protein catabolism, and DNA damage repair, respectively. 5–10 larval brains of each genotype were dissected and observed.

# Results

## Nuclear Pros as a marker for NB early differentiation and regulated by DNA damage repair genes

Previous studies have documented that the disturbance of NB proliferation induces nuclear Pros accumulation at the third instar larvae stage in *Drosophila*, leading to decreased numbers of NBs and premature differentiation (Wu et al, 2019; Sood et al, 2021). Therefore, using nuclear Pros as a marker for premature NB differentiation, we screened for genes involved in NB premature termination versus normal proliferation. We selected 300 genes that had been previously identified to function in the process of transcriptional regulation, cytoplasmic transportation, and genomic instability during the self-renewal and differentiation of NBs (Neumuller et al, 2011; Wu et al, 2019) and performed an initial RNAi screening using the NB-specific expression driver *worniu*-GAL4 and UAS-Dicer2. Third instar larval brains were then dissected and immune-stained with the NB marker Dpn and Pros antibodies. Confocal microscopy images showed that the knockdown of many candidate genes, for example, Cullin-2 (Cul2) and Bloom syndrome gene (BLM), could cause a nuclear Pros phenotype in NBs, associated with decreased NB number in the brain (Fig 1A–C"). Based on GO analysis on their biological functions, these genes were sorted into four categories related to nucleocytoplasmic transportation, protein catabolism, transcription, or DNA damage repair (Table 1). We focused on the genes involved in the DNA damage repair process. We then performed a second RNAi screening combined with single doses of X-ray treatment to investigate whether the nuclear Pros phenotype of NBs was related to these DNA damage repair pathway genes under specific IR stress. We collected as many RNAi lines as we could for each gene to exclude the possible off-target effect of RNAi and to avoid background noise in our experiments (Fig S1). Results showed that simply under RNAi

**Table 2. RNAi screening for genes involved in DNA damage repair pathways in NBs.**

| Gene | Worniu–Gal4 derived | | Percentage of NBs with prosnuclei | Significance |
|------|-----------|-----------|-----------------------------------|--------------|
|      | untreated | 30 Gy X-ray |                                 |              |
| $w^{1118}$ | No | Yes | 8.6% | |
| ATM(tefu) | No | Yes | 10.1% | ns |
| ATR(mei-41) | No | Yes | 14.3% | ++ |
| ATM(tefu); ATR(mei-41) | No | Yes | 17.8% | ++ |
| CtlP | No | Yes | 11.3% | + |
| Ku80 | No | Yes | 10.0% | ns |
| Mre11 | Yes | Yes | 10.9% | + |
| Nbs | Yes | Yes | 12.6% | ++ |
| Rad50 | No | Yes | 11.9% | + |
| BRCA2 | No | Yes | 15.4% | ++ |
| RAD51(spn-A) | No | Yes | 11.2% | ++ |
| Blm | Yes | Yes | 4.9% | + |
| WRNexo | Yes | Yes | 10.0% | - |
| top3a | No | Yes | 10.0% | ns |
| Chk1 | No | Yes | 10.0% | ns |
| Chk2 | No | Yes | 10.0% | ns |
| mre11/Chk1 | Yes | Yes | 10.0% | + |
| mre11/Chk2 | Yes | Yes | 10.0% | ns |
| p53 | No | Yes | 30.0% | +++ |

By using neural system specific expression of GAL4 knockdown essential genes that relate to the homologous recombination pathway or the non-homologous end-joining pathway. 5–10 larval brains of each genotype were dissected and observed.

knockdown only some cases showed nuclear Pros in NBs (Fig S1 and Table 2), whereas when combined with IR treatment, all knockdown of genes (Table 2) showed a nuclear Pros phenotype.

**X-ray irradiation triggers DNA damage repair and nuclear Pros in NBs**

Given the phenotype of nuclear Pros in NBs upon the knockdown of DNA damage repair genes, we hypothesized that these genes were susceptible to DNA damage stress, or particularly required for NB maintenance, under conditions of environmental stress such as irradiation. To investigate the mechanisms involved in DNA damage repair in neural stem cells, we first exposed *Drosophila* larvae to excessive amounts of X-ray radiation to trigger DNA damage in WT larva brains. We found that exposure to X-ray at a dose of 30 Gy in the early third instar larvae (48 h after egg lay [AEL]) caused pupal lethality with the larval brain morphology showing corresponding shrinkage (Fig 2B). We noticed that 10% of the NBs presented nuclear Pros under IR conditions (Fig 2C–F'). We performed co-immunostaining of the DNA damage marker γH2Av and the NB marker Asense (Ase), and observed an obvious accumulation of γH2Av-positive foci in the NBs of the brain upon IR treatment, in contrast to that of controls (Fig 2G–I), indicating that DNA damage had occurred after IR treatment. These results suggest that X-ray irradiation–induced nuclear Pros could be considered as a parallel event to that of the DNA damage repair process in NBs.

**X-ray IR induces premature differentiation in NBs**

We then used the different X-ray dosages and exposure durations causing nuclear Pros phenotypes in NBs at different developmental stages of larval brains. With a dose of 30 Gy X-ray irradiation (IR) exposure for 11 min at the stage of 48 h AEL, we dissected the larval brains at 2 h, 12 h, 24 h or 48 h points after IR treatment and examined the phenotype of NBs with nuclear Pros in both experimental groups and control groups. Results showed that at the 24 h-point, the brain presented with the highest number of NBs showing nuclear Pros, beyond which, at the 48 h-point, both the total number of NBs (Fig 3A, right) and the percentage NBs with nuclear Pros had decreased (Fig 3B, right). The control groups do not show these effects in either the NBs number (Fig 3A, left) or the Pros phenotype (Fig 3B, left). This suggests that the NB response to environmental stress builds slowly, peaking at the 24 h-point, beyond which the NB phenotype is reduced. Upon further extending the exposure time to 25 min and then to 33 min (Fig S2A–D), no further significant differences in NBs phenotype were noted compared with that of the 11 min exposure (Fig S2E). This indicates that the proportion of NBs affected remains the same regardless of X-ray irradiation exposure duration. We therefore used the combination of the larvae at 48 h AEL with IR treatment for 11 min as a regular experimental condition.

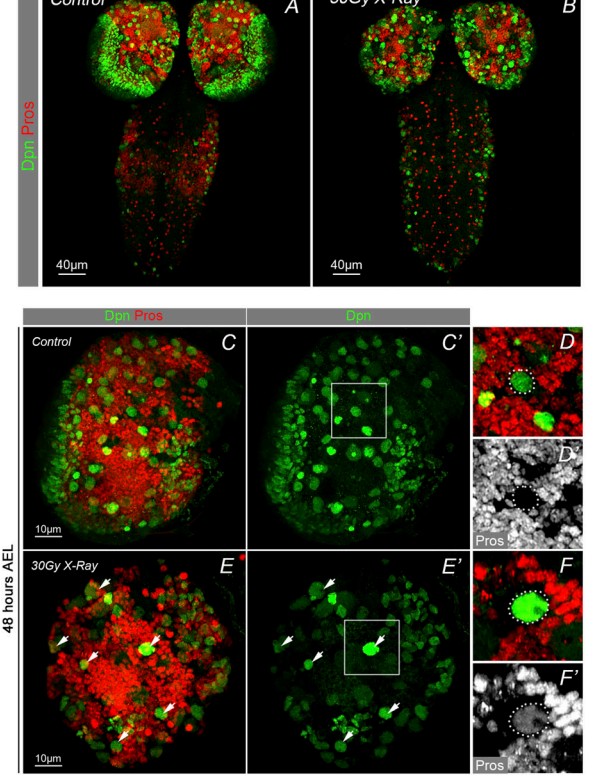

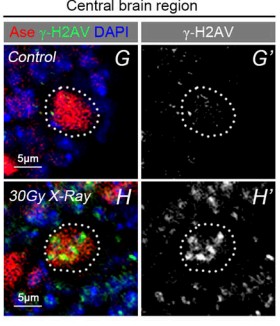

Central brain region

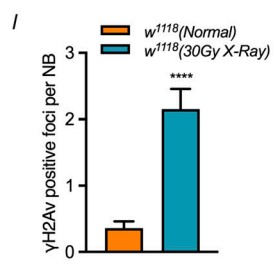

**Figure 2. X-ray irradiation triggers DNA damage repair and nuclear Pros in NBs.**
**(A, B)** Early second instar larval exposure to high dosage of X-ray irradiation. Treatment at pupal stage leads to Drosophila pupa lethality. Dissection of the brain of third instar larvae showed morphological shrinkage of larval brain and significant phenotype of Pros accumulation in the neuroblasts beyond exposure. **(C, D')** Early instar larval brain staining with neuroblast marker Dpn (green) and transcriptional factor Pros (red). **(E, F')** WT early instar larval treatment with 30 Gy X-ay and staining larval brain with stem cell marker Dpn (green) and transcriptional factor Pros (red). High dosage X-ray treatment induced significant Pros accumulation in the nucleus of NBs. **(G, H)** immunostaining of DNA damage marker γH2Av suggest that high dosage of X-rays could induce significant DNA damage and γH2Av positive foci in the nucleus of larval neuroblasts. 5–10 larval brains of each genotype were dissected and observed. **(I)** Statistical measure of γH2Av in neuroblasts show a dramatic increase of DNA damage compared with the non-treatment control. Quantified NB number of 30 Gy-X-ray treatment group, n = 68; quantified NB number of control group, n = 136. Statistic data in (I) are presented as mean ± SD, *P < 0.05, **P < 0.01, ***P < 0.001, ****P < 0.0001; unpaired t test.

Previous studies have shown that apoptosis occurs in cultured human multipotent mesenchymal stromal cells under IR treatment (Ruhle et al, 2018), and that such a DNA damage agent subjected to third instar *Drosophila* larvae can lead to impairments in later developmental survival and motor function in adult flies(Sudmeier et al, 2015). We performed immunostaining with the apoptosis marker anti-caspase 3 in the larval brains at 48 h AEL with X-ray exposure for 11 min. Results showed that there was no significant difference in the apoptosis signals between the X-ray irradiated brains and control counterparts (Fig 3C and D'), indicating that no apoptosis had been initiated. As cell cycle exit is the consequence of the nuclear accumulation of Pros, this also indicates that cells that are likely undergoing differentiation (Li & Vaessin, 2000; Chia et al, 2001; Maurange et al, 2008). We performed mitotic marker phospho-histone 3 (PH3) staining (Figs 3E and 4F'') and found that there were less PH3-positive cell in the brains with IR treatment, compared with the controls (Fig 3G). These observations suggest that irradiation treatment had forced cell cycle exit, terminated NB cell fate, and triggered NBs to undergo premature differentiation. We conducted the experiments with EdU pulse in the brains at 24 h-point and 48 h-point after IR exposure Fig S3A–C'). As shown in Fig S3G, the EdU signals showed significant decreases, compared with the controls. This suggests that under IR condition, as some of NBs terminated their cell fates earlier by undergoing premature differentiation, the EdU-labeled postmitotic cells correspondingly reduced.

**The MRN complex and the HR repair pathway prevent nuclear Pros localization in NBs under IR stress**

As X-ray IR–induced DNA damage to the *Drosophila* central nervous system and activated the DNA damage repair process in NBs, we sought to figure out to what extent the typical NHEJ repair pathway and the HR pathway participate in regulating NB cell fate. Focusing on the main regulatory genes of these two distinct pathways, we performed a third screening to evaluate the role of NHEJ and HR pathways related to the NBs with nuclear Pros phenotype in response to IR treatment.

The NHEJ pathway is the main pathway responsible for repair of DSBs and operates throughout the entire cell cycle. In this, the Ku70/Ku80 heterodimer complex forms on the broken DNA ends to recruit and activate downstream protein kinases, such as DNA-PKcs, LIG4, and XRCC4, to complete the repair process (Marini et al, 2019). As shown in Table 2, we noticed that when knocking down each of these NHEJ orthologue genes specifically in NBs, including Ku80 or CtIP knocking down under the IR treatment result in insignificant nuclear Pros phenotype compared with WT. Only Ku80 knockdown caused the nuclear Pros phenotype in NBs under IR treatment (Fig S4A–D).

The HR pathway is an error-free repair system operating only in the S and G2 phases of the cell cycle (Jasin & Rothstein, 2013). It involves DNA breakage ends that go through resection by a heterotrimer of the MRE11 complex, with single-strand DNA at double-strand breaks processed by exonucleases to form long single-strand

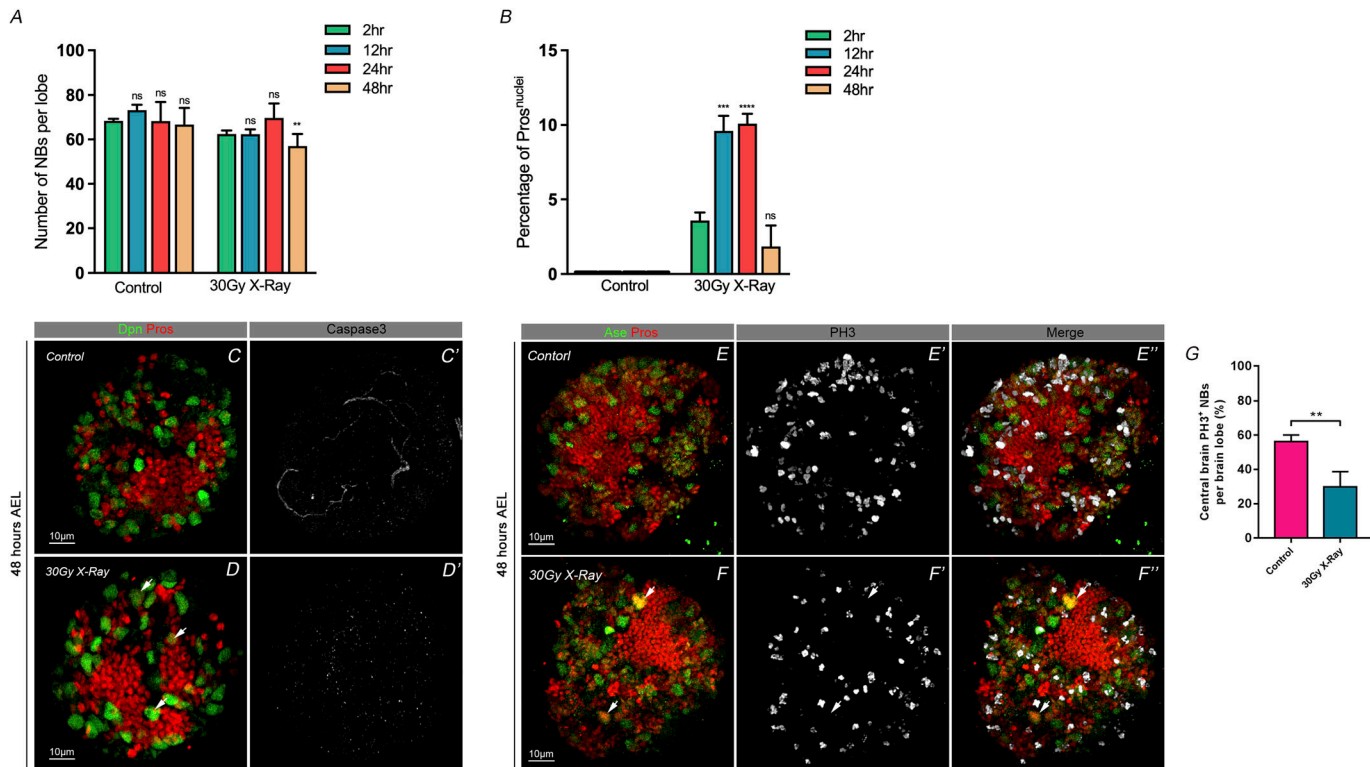

**Figure 3. X-ray irradiation induces NB premature differentiation.**
**(A)** Using high dosage of X-ray irradiation (30 Gy) treatment of larvae 48 h after egg lay, compared with control group under the normal conditions revealed a reduction of neuroblasts at 48 h after exposure. **(B)** The percentage of NB nuclei with Pros had increased significantly from 12–24 h after exposure to X-ray of 30 Gy. This ratio then had decreased by 48 h after irradiation. The phenotype may be due to the rapid reduction of neuroblast number. **(C–C', D–D')** Apoptosis marker caspase 3 staining shows non-significant changes between control group and X-ray–treated group. **(E–E', F–F')** Staining of mitotic marker phosphor-Histone 3 showed cell cycle arrest after 30 Gy irradiation. **(G)** Statistic of phosphor-histone three positive loci between control group and X-ray–treated group. Statistic data in (A, B) are presented as mean ± SD, *$P < 0.05$, **$P < 0.01$, ***$P < 0.001$, ****$P < 0.0001$; one-way ANOVA.

DNA. Recombinase RAD51 then cooperates with BRCA1 and BRCA2 to perform a homology search for sister chromatids and to generate double Holliday junctions. This, in turn results in downstream products or promotes a through synthesis-dependent strand annealing pathway that newly synthesize strand anneals on the other side of the original breaks (Ma et al, 2017). As Rad51 is a marker for the HR repair pathway (van Wijk et al, 2022), and that the MRN complex (consisting of Mre11, Rad50, and Nbs1) is noted to activate the HR pathway in response to DSBs when the NHEJ pathway is nonfunctional (Jackson & Bartek, 2009; Polo & Jackson, 2011; Reginato & Cejka, 2020). By contrast, knockdown of all of the HR repair pathway genes, including Mre11, RAD50, Nbs, RAD51/spn-A, Chk1/Chk2/mre, BRCA2, WRNexo, and Blm, resulted in the nuclear Pros phenotype in NBs under IR condition (Table 2 and Fig S5). This suggests that alterations in the HR repair pathway may have a dominant effect on the NB fate under IR stress over those of NHEJ.

We tested the respective suppression of Rad51 and the subunits of the MRN complex, Mre11, Rad50, Nbs, when combined with IR treatment (Fig 4A–E). Results showed that under IR condition, knockdown any of these genes led to an increased numbers of NBs with nuclear Pros, compared with the WT (Fig 5F). These data suggested that the MRN complex and HR pathway play critical roles in regulating the NB cell fate under IR stress (Fig 5G).

## The DNA damage sensor ATR/*mei-41* is involved in the maintenance of NBs

The main DNA damage repair transducer is ATM serine/threonine kinase (ATM), which operates at the DSBs and at ataxia–telangiectasia and Rad3 related (ATR). Activation of ATM and ATR can then phosphorylate various substrates include the DNA damage indicator H2A histone family member X, and activate both CHK1 and CHK2 (Awasthi et al, 2015). We examined the effects of suppressing DNA damage sensors on the nuclear Pros in NBs, (Fig 5A–D'), and found that the knockdown of the orthologue gene ATR/*mei-41*, rather than ATM/*tefu*, in NBs, combined with the treatment of X-ray exposure caused a significant increase in the number of NBs with nuclear Pros, compared with the controls (Fig 5E). These data suggest that *mei-41* rather than *tefu*, may have distinct regulatory relevance to the stem cell fate termination process.

## *mei-41*–mediated nuclear Pros under IR stress is dependent on WRNexo

WRN contains an RecQ C-terminal domain and a helicase and ribonuclease D C-terminal (HRDC) domain. These are largely responsible for DNA and protein binding. The RecQ family of helicases is known as the "guardians of the genome" because of their roles in DNA replication, repair, and maintenance of genomic integrity. Like

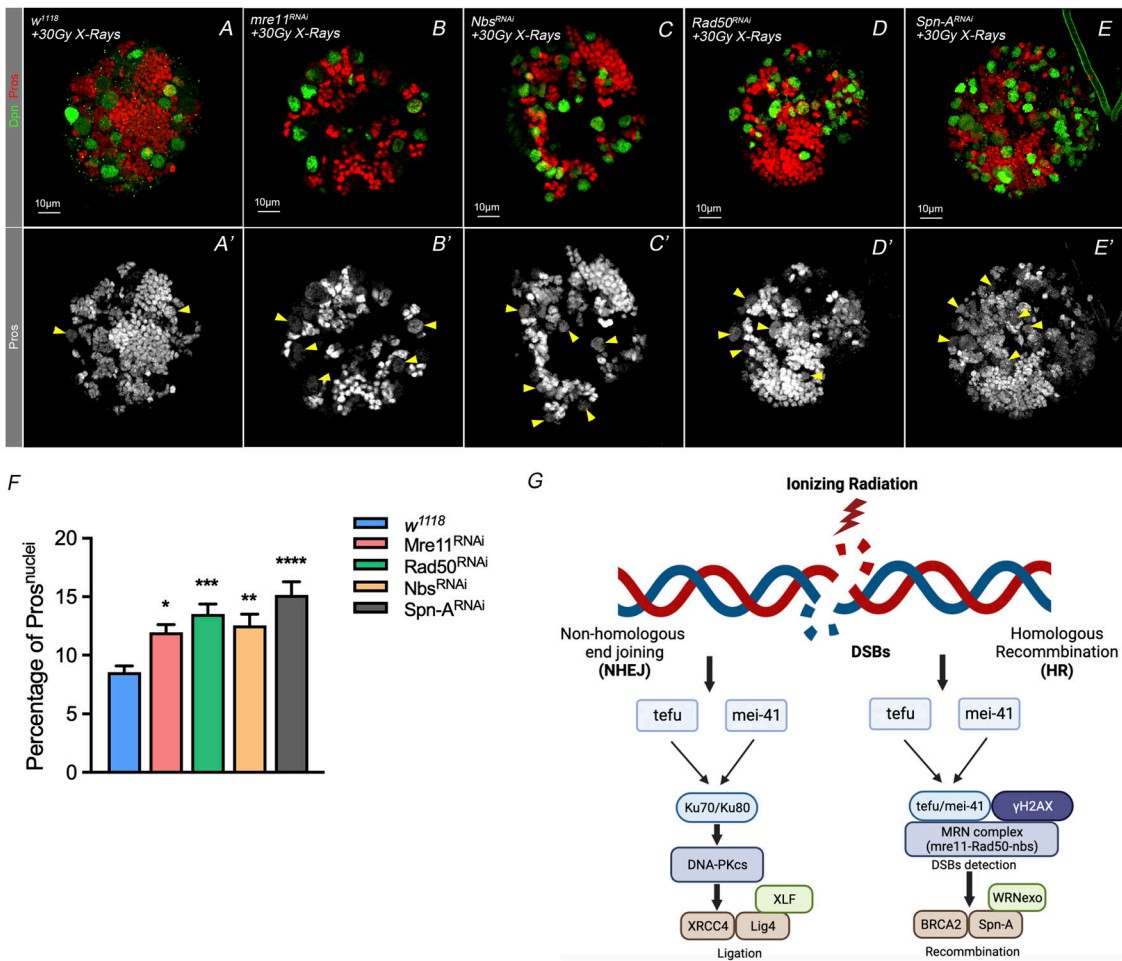

**Figure 4. The MRN complex and the HR repair pathway prevents nuclei Pros accumulation in NBs after IR stress.**
**(A, B, C, D')** The MRN complex, consisting of mre11, Nbs, and Rad50, functions in the resection process when DSB occurs. Down-regulation of either of its subunits causes an increase of the nuclear localized pros ratio under the DSB after exposure to X-ray. Rad51 promotes homologous recombination in damaged cells and is considered as a marker for determining the regulation of the homologous recombination process after the condition of double-strand breaks. **(E–E')** Specific knocking down of Spn-A, the *Drosophila* homolog of Rad51, in neuroblasts results in significant elevation of the nuclear Pros phenotype beyond treatment with a high dosage of X-ray irradiation. **(F)** Statistic of nuclei with Pros after treatment with 30 Gy X-ray. **(G)** Schematic model of the role of the homologous recombination repair and the non-homologous end-joining pathway in the *Drosophila* DNA damage repair process in NBs is involved in promoting premature nuclear Pros, which is evidenced from the knockdown of the ATR/ATR upstream, through the MRN complex in DSB detection, to the regulatory proteins in the recombination process. Statistic data in (F) were presented as mean ± SD, *P < 0.05, **P < 0.01, ***P < 0.001, ****P < 0.0001; 0ne-way ANOVA.

other RecQ family members, WRN exhibits ATP- dependent DNA helicase activity. In *Drosophila*, the WRNexo gene encodes a protein with 35% identity and 59% similarity to the exonuclease domain of human WRN. Purified WRNexo exhibits exonuclease activity on single-strand DNA, double-strand DNA with 5' overhangs, and substrates representing replication bubbles (Shen & Loeb, 2000b; Bolterstein et al, 2014). We performed WRNexo knockdown in NBs and found that this reversed the phenotype of NBs after treatment with X-ray IR, decreased the numbers of NBs with nuclear Pros (Fig 6C–C' and F). We conducted the experiments with EdU pulse in the WRNexo RNAi brains at 24 h-point and 48 h-point after IR exposure (Fig S3D–F'). As shown in Fig S3G (right), the EdU signals showed no difference at 24 h-point and 48 h-point after IR exposure, suggesting activated HR pathway postponed premature differentiation.

We then carried out a double knockdown of WRNexo and the key genes for DNA damage sense and transition to HR repair pathways,

*mre11* and *mei 41* (Fig 6A–E'), and found that the ratio of NBs with nuclear Pros was decreased in the double knockdown of these two, as compared with one singly, when combined with X-ray irradiation (Fig 6F). We concluded that WRNexo RNAi knockdown in NB caused hyper-recombination activity and rescued the phenotype caused by the combination of X-ray and other HR component knockdown.

## Discussion

During early development stages, stimulation of the metabolism, specific genetic background, and/or physical or chemical factors, can all cause the accumulation of damage to the genome. DNA damage repair defects can then lead to alterations of stem cell fate (Sang et al, 2022; Schimenti et al, 2022). Here, we show that a high

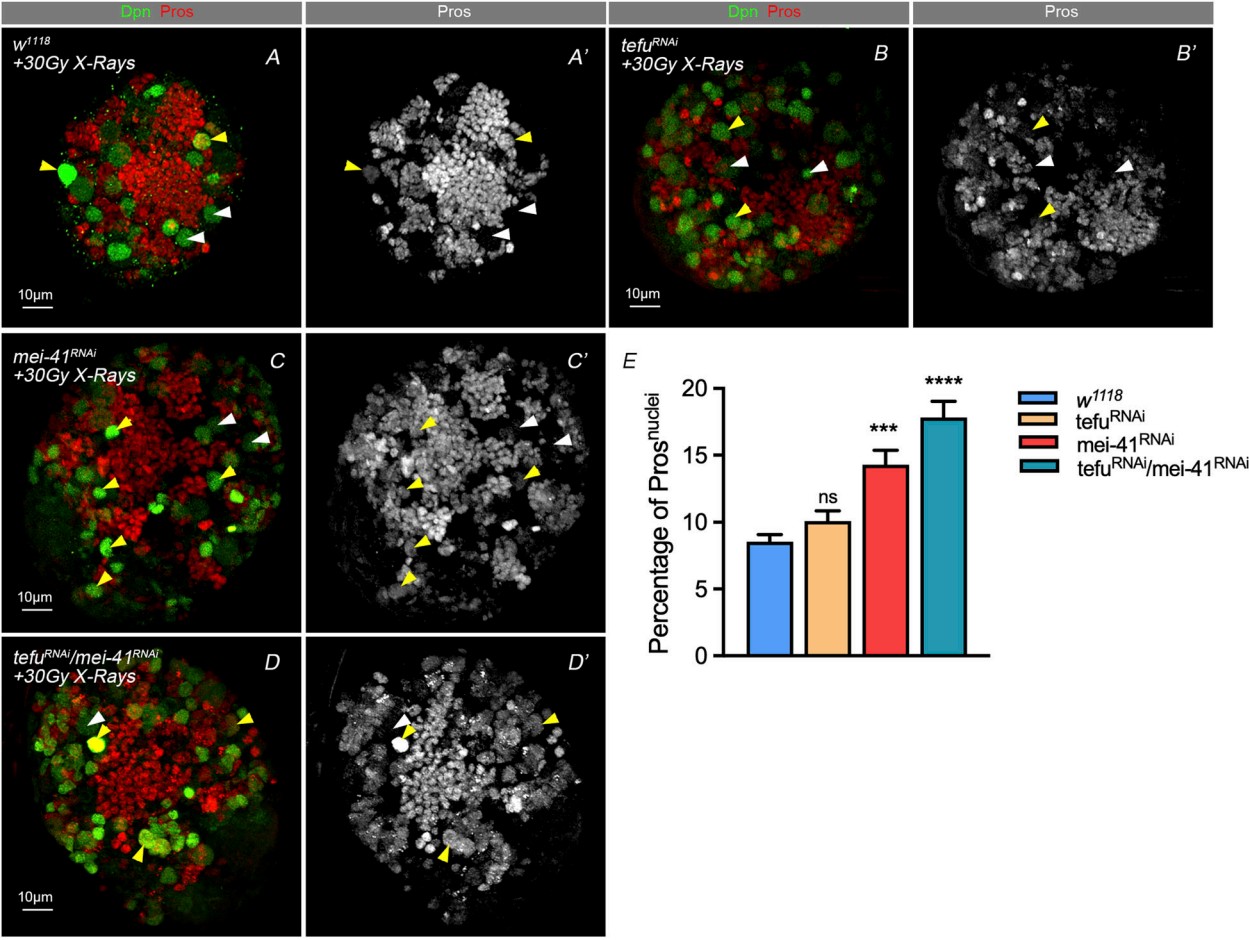

**Figure 5. The DNA damage sensor ATR/*mei-41* is involved in NB fate termination.**
**(A, B, C')** ATM/ATR is considered as a sensor for the DNA damage response. Down-regulation of either of its homologous genes in *Drosophila* causes an increase of neuroblasts with nuclear-localized Pros. **(D–D')** IF staining revealed that double knockdown resulted in a far more significant phenotype compared with single component down-regulation. **(E)** Statistic of nuclei with Pros after treatment with 30 Gy X-ray. Statistic data in (E) are presented as mean ± SD, *$P$ < 0.05, **$P$ < 0.01, ***$P$ < 0.001, ****$P$ < 0.0001; one-way ANOVA.

dosage of X-ray IR induces DNA damage, in which the active DNA damage repair process is associated with nuclear Pros accumulation in NBs, resulting in premature differentiation and cell fate termination. Along with the DNA damage sensing and repair pathways, it is the MRN complex and HR pathway that play dominant roles in this process. The DNA damage sensor ATR/mei-41 also participates in regulating IR-induced nuclear Pros, which itself is dependent on the function WRNexo.

Previous study has reported that mutation of Drosophila tefu/ATM leads to neuronal and glial cell death in the adult brain (Petersen et al, 2012). We therefore considered it possible that tefu/ATM knockdown could induce the apoptosis of NBs. However, as shown in Fig 4 and in the observations of our previous study (Wu et al, 2019), the neclear Pros of NBs phenotype does not relate directly to apoptosis. In Fig 5, we show that the statistical results of the knockdown of tefu alone are not significantly different from those of the knock down of mei-41 (the homolog of ATR). However, there is a significant nucleation phenotype under the double knockdown of both mei-41 and tefu, compared with the phenotype

of the knocking down each of them singly. This may be due to a redundancy and also a dominant effect of mei-41.

WRNexo has been reported to be involved in many aspects of DNA metabolism, including DNA replication, recombination, repair, transcription, and telomere maintenance (Bolterstein et al, 2014). In *Drosophila*, two WRNexo mutants have already been reported, in which WRNexoe04496 (Bolterstein et al, 2014), causes a severe reduction in WRNexo expression, resulted from the presence of a piggyBac {RB} transposable element in the 5'-UTR of WRNexo. WRNexoe04496 flies exhibit high sensitivity to the topoisomerase I inhibitor and display hyper-recombination (Bolterstein et al, 2014). A second mutant, WRNexoD229V (Shen & Loeb, 2000a) contains a point mutation that ablates exonuclease activity at physiological temperatures. Like WRNexoe04496, WRNexoD229V mutants display hyper-recombination. We hypothesize that WRNexo may function downstream of the DNA sensory process and transduce signals to activate the nuclear Pros accumulation process, which then prematurely regulates neural stem cells to terminate their stem cell fate rather than continue proliferation.

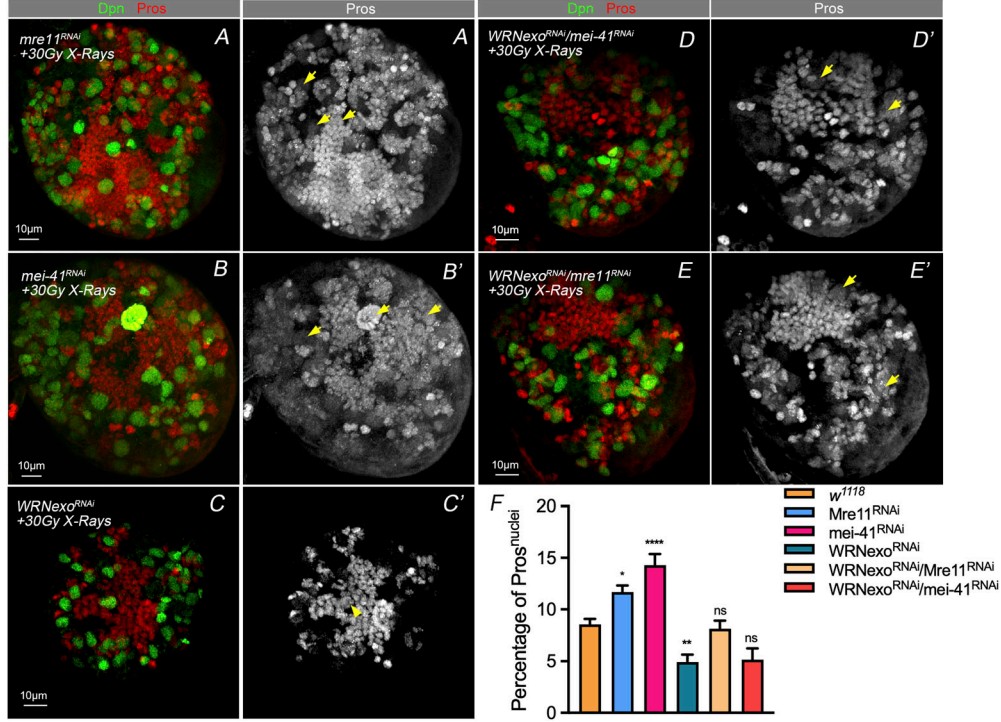

**Figure 6. *mei-41*–mediated nuclear Pros after IR stress is dependent on WRNexo.**
**(A, B, C, D, E')** A second round of screening found that knockdown of WRNexo could prevent the nuclear Pros in the brains of *mei-41* knockdown NBs after IR stress. IF staining shows it could also perfectly rescue the phenotype caused by other DSB-related genes. **(C)** Statistic of nuclei with Pros after treating with 30 Gy X-ray. Statistic data in (F) were presented as mean ± SD, *$P < 0.05$, **$P < 0.01$, ***$P < 0.001$, ****$P < 0.0001$; one-way ANOVA.

During NB asymmetric divisions, Pros, as a NB cell fate determinant, is localized in the cytoplasm of NB and is segregated exclusively into GMCs during NB asymmetric divisions. Regardless of whether Pros either prematurely enters the nucleus of NBs at the larval stage or normally enters at the pupa stage, NBs then exit the cell cycle and terminate their NB cell fate. No mitotic NBs are then detected in the central brain or ventral nerve cord of adult flies (Wu et al, 2019). Under IR stress, DNA damage repair defects caused early entry of Pros into the nucleus of NBs, which could lead to cell cycle exit and termination of NBs asymmetric division.

In our present study, the dysfunction of DNA damage repair (by RNAi of DNA repair related genes or X-ray IR) induces Pros to enter the NBs nucleus which corresponds to a reduction of NB population. This suggests that although the DNA damage repair in neural stem cells is irreversible beyond such IR exposure, the stem cells can prevent the production of mutant cells by facilitating NB cell fate termination via regulating the entry of the transcription factor Pros into the nucleus prematurely. This process of eliminating mutant stem cells is similar to the later natural extinction of NBs at the pupal stage, indicating that the endogenous stem cell termination mechanism is also involved in the early termination of NB fate induced by X-ray irradiation. This is quite different from the mechanism by which differentiated cells often choose programmed cell death or apoptosis to terminate their cell fates under external stress (Kroemer et al, 2022; Zhu et al, 2022). Here, we demonstrate an activation pathway that instead regulates an intrinsic termination mechanism of stem cells and show that this activation pathway has the function of monitoring and clearing mutant stem cells. Pros itself is highly evolutionarily conserved where PROX1, an orthologue protein in

other insects and vertebrates, has had similar functions noted (Oliver et al, 1993; Elsir et al, 2012). Comparable activation pathways and intrinsic stem cell termination mechanisms are also likely to be present in other organisms, including mammals, facilitating the elimination of dysfunctional stem cells to decrease the likely incidence of tumorigenesis.

In our primary RNAi screening for NBs with nuclear Pros, we identified the DNA repair genes involved in regulating nuclear Pros in NBs. This promoted us to establish an IR stress model to confirm if DNA damage might induce nuclear Pros in larval brain NBs. Our secondary RNAi screening targeting DNA damage repair–related genes further supported this result. Moreover, the X-ray–irradiated NBs presented significant DNA damage signals and showed increased numbers of NBs with nuclear Pros. These results suggest that if the extent of DNA damage exceeds the endogenous repair capacity (or where the endogenous repair pathway is disturbed or the DNA damage dosage is increased) the NBs may respond by initiating a nuclear Pros accumulation process to avoid propagating such a dysfunction. Along with the shrinkage of larval brains, the NB number was decreased at 48 h after irradiation exposure. However, caspase 3 staining suggested that despite apoptosis having not occurred, the NBs had been prematurely terminated. Along with the observation of a high percentage of NBs with nuclear Pros noted at 24 h, AEL expression also indicated premature NB differentiation. PH3 staining also confirmed that NBs had exited the cell cycle after IR stress.

In conclusion, our study reveals that the DNA damage sensor ATR, the MNR complex, and the HR repair pathway acts to regulate the maintenance of NBs. We have demonstrated that NBs undergo the premature differentiation (terminal division) via an accumulation of nuclear Pros under IR stress. It is this premature

differentiation, rather than apoptotic cell death that provides NB with resistance against the consequences of stress, resulting in inappropriate, undesired, and potentially uncontrolled propagation. A similar mechanism may occur in other organisms including humans relating to termination of mutant stem cells to prevent tumor initiation under IR or other stress. Exploring this emerging mechanism may shed light on the understanding of the mechanisms involved in brain tumor formation under environmental stress.

## Materials and Methods

### Fly stocks and genetics

All flies were maintained at 25°C. The *Drosophila* stocks (and their sources) used in this study include: $w^{1118}$, UAS-*Dicer2*; *wor*-GAL4, *tefu*$^{RNAi}$ (THU5591 and THU5787; Tsinghua Drosophila Center), mei-41$^{RNAi}$ (THU0256 and THU5335; Tsinghua Drosophila Center), *Top3-α*$^{RNAi}$ (TH04206.N; Tsinghua Drosophila Center), *grp*$^{RNAi}$ (THU2601 and THU5194; Tsinghua Drosophila Center), *Dmnk*$^{RNAi}$ (TH01867.N and THU0019; Tsinghua Drosophila Center), *Dmp53*$^{RNAi}$ (THU2533, THU4578, THU4625, THU5318; Tsinghua Drosophila Center), *string*$^{RNAi}$ (THU0678, THU4503; Tsinghua Drosophila Center), *14-3-3ε*$^{RNAi}$ (THU0336 and THU4849; Tsinghua Drosophila Center), *14-3-3ζ*$^{RNAi}$ (THU4651 and THU4850; Tsinghua Drosophila Center), *CycE*$^{RNAi}$ (THU4501; Tsinghua Drosophila Center), *mre-11*$^{RNAi}$ (THU5229 and TH01614.N; Tsinghua Drosophila Center), *Spn-A*$^{RNAi}$ (THU5088 and TH02208.N; Tsinghua Drosophila Center), *WRNexo*$^{RNAi}$ (THU3811; Tsinghua Drosophila Center), *Nbs1*$^{RNAi}$ (65971; BDSC). Constructs were injected at the Core Facility of Drosophila Resource and Technology, CEMCS, CAS, following standard methods.

### X-ray treatment assay

Flies were maintained in cages and the new-born embryos (2–3 h) were collected into dishes containing regular corn food meal medium. The collected embryos were maintained at 25°C for 48 h. The resulting larvae were transferred onto a new plate only containing agar, and treated with 30 Gy of X-ray for 10 min. The larvae were then transferred back to a regular corn food meal medium and maintained for a further 24 h. Brains were then dissected for immunostaining.

### Immunofluorescence staining and antibodies

Different stages of larval brains and adult brains were dissected in ice-cold Schneider's *Drosophila* medium (Gibco). Samples were fixed for 18 min in PBS (10 mM NaH2PO4/Na2HPO4, 175 mM NaCl, pH 7.4) with 4% paraformaldehyde at room temperature (Zhang et al, 2016). The samples were incubated with primary antibodies at 4°C overnight, and then with secondary antibodies at room temperature for 1–2 h. Antifade mounting medium (P0126; Beyotime) was used to protect the fluorescent signals of the samples. Images were obtained using an Olympus FV1000 confocal microscope and processed using Adobe Photoshop.

The primary antibodies in this study were as follows: mouse anti-Pros (1:50; DSHB); guinea pig anti-Dpn (1:1,000, a gift from Y. Cai); rabbit anti-Ase (1:1,000, a gift from Y. Cai); rabbit anti-caspase 3 (1:1,000, Asp175; Cell Signaling); rabbit anti-phospho-histone 3 (Ser10) (1:1,000; Millipore); rabbit anti-histone H2AvD pS137 (1:1,000; Rockland). All commercial secondary antibodies used were from the Jackson Laboratory. DNA was stained with DAPI (C1002; Beyotime) at 1:2,000.

### Edu staining

Larval brains were dissected in ice-cold Schneider's Drosophila medium (Gibco). EdU-labeling solution were prepared by diluting the EdU reagent (EdU-594,C7800S; BeyoClick)in Schneider's Drosophila medium. Samples were added with the EdU-labeling solution and incubated at room temperature for 30 min, and then fixed with 4% paraformaldehyde for 15 min. Using PBS solution the sample were washed for 3–5 min for three times. The samples were permeabilized using permeabilization solution (PBS solution with 0.3% Triton X-100) for 10–15 min and then washed with PBS for three times. The detection solution was prepared using click reaction buffer precisely for 15 min before detection, adding detection solution to the sample and incubating for typically 30 min. Before observation by the confocal microscope, the samples were washed with 3% PBS-T for three times and incubated in anti-fade mounting medium.

### Statistical analysis

All data are expressed as the mean ± SD. Statistical data were processed using unpaired two-tailed $t$ test, and one-way ANOVA to test for differences between two or more groups of data in GraphPad Prism to determine if there are significant differences in the mean values of different groups.

## Data Availability

The online version of this article (website) contains supplementary material, which is available to authorized users.

## Supplementary Information

## Acknowledgements

We thank Yan Song and Tsinghua Stock Center and Bloomington Stock Center (BDSC) for the fly stocks and the Core Facility of Drosophila Resource and Technology of CAS for fly microinjections. This study was supported by grants from National Key R&D Program of China (2018YFC1004900).

## Author Contributions

X Xu: conceptualization, data curation, software, formal analysis, validation, investigation, visualization, methodology, and writing—original draft.

H An: investigation, visualization, methodology, and writing—original draft.

C Wu: data curation, formal analysis, investigation, and visualization.

R Sang: conceptualization, data curation, formal analysis, investigation, visualization, and methodology.

L Wu: conceptualization, software, formal analysis, investigation, and visualization.

Y Lou: formal analysis, investigation, and visualization.

X Yang: conceptualization, funding acquisition, methodology, and writing—review and editing.

Y Xi: conceptualization, resources, data curation, supervision, funding acquisition, validation, investigation, methodology, project administration, and writing—original draft, review, and editing.

## Conflict of Interest Statement

The authors declare that they have no conflict of interest.

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
