## [Reviewer comments · Life Science Alliance]

Life Science Alliance

HR repair pathway plays a crucial role in maintaining neural stem cell fate under irradiation stress

Xiao Xu, Huanping An, Chen Wu, Rong Sang, Litao Wu, Yuhan Lou, Xiaohang Yang, and Yongmei Xi

DOI: <https://doi.org/10.26508/lsa.202201802>

Corresponding author(s): Yongmei Xi, Women's Hospital, School of Medicine, Zhejiang University and Xiaohang Yang, Zhejiang University

Review Timeline:

Submission Date:	2022-11-04
Editorial Decision:	2022-12-14
Revision Received:	2023-03-16
Editorial Decision:	2023-04-11
Revision Received:	2023-04-26
Editorial Decision:	2023-04-28
Revision Received:	2023-05-03
Accepted:	2023-05-04

Scientific Editor: Novella Guidi

Transaction Report:

December 14, 2022

Re: Life Science Alliance manuscript #LSA-2022-01802-T

Dr. Yongmei Xi
Zhejiang University
Institute of Genetics, Zhejiang University and Department of Genetics, Zhejiang University School of Medicine
Yuhangtang Road 866, Xihu District
Xihu district
Hangzhou, Zhejiang 310058
China

Dear Dr. Xi,

Thank you for submitting your manuscript entitled "MRN complex and the HR repair pathway dominate in maintaining the fate of neural stem cells" to Life Science Alliance. The manuscript was assessed by expert reviewers, whose comments are appended to this letter. We invite you to submit a revised manuscript addressing the Reviewer comments.

Thank you for this interesting contribution to Life Science Alliance. We are looking forward to receiving your revised manuscript.

Sincerely,

B. MANUSCRIPT ORGANIZATION AND FORMATTING:

Reviewer #1 (Comments to the Authors (Required)):

In their manuscript entitled "MRN complex and the HR repair pathway dominate in maintaining the fate of neural stem cells", Xu et al. describe the activation of DNA repair pathways in neuroblasts (NBs) of the developing *Drosophila* brain as a consequence of exposure to X-ray irradiation (IR).

The authors show a correlation between IR-induced nuclear accumulation of Prospero and loss NB numbers. Thus, Xu et al. speculate that IR induces premature differentiation of NBs. They suggest that premature terminal differentiation represents a preventive cellular mechanism that protects the organism from retention of damaged or mutant stem cells which might otherwise develop into tumor cells.

Knock-down experiments using RNAi to reduce the expression of several DNA damage repair genes result in increased numbers of NBs showing nuclear accumulation of Prospero and an overall decrease of NB numbers, suggesting that NBs undergo terminal differentiation and are thus lost from the stem cell pool. The authors conclude that the DNA repair pathway plays an important role in the clearance of IR-induced DNA damage and subsequently in the prevention of premature NB differentiation. Using further RNAi screens, Xu et al. uncover that DNA repair occurs in an ATR- and WRNexo-dependent manner.

Major comments:

The manuscript shows a lot of inconsistencies, labelling errors and confusing figure presentation. Semantically, the text often infers causal relationship while at best the authors show correlations. I strongly invite the authors to adjust their writing style and claims to the content of their data. Given several other published reports showing a role of IR in inducing premature NB differentiation I suggest further experiments to lift the current manuscript to a higher level of novelty.

1. Inconsistencies:

- a. The main text describing results of figure 2 (line 138) claims that "each of these" genes result in nuclear Pros. However, the images and also the legend of Figure 2 show that only two of the tested genes result in nuclear Pros. Please correct the text in line 138 accordingly.
- b. In many parts of the manuscript, the authors state that *tefu* knock-down does not result in significantly increased nuclear Pros and premature NB differentiation (e.g. Figure 2, Table 2). However, in Figure 6 they show a significant increase of nuclear Pros in *tefu* knock-down brains. Please elaborate on these contrary data and correct the data display or interpretation.
- c. Figure 3 panels G-H' do not have a DAPI signal in the demarcated nuclei. Please elaborate why there is no DAPI and how you determined that the respective demarcated area is a nucleus if there is no DAPI signal?
- d. Figure 4, panels A-B: the control brain shown in A contains more nuclear Pros positive NBs than the irradiated brain shown in B. How does this fit with your data and the main message of your manuscript? Please replace the image shown in A with a more representative image. However, if A is representative you might want to rethink the main message of the paper. Similarly, images shown in Figure 3 panels C and E show similar amount of nuclear Pros positive NBs to me.
- e. Figure 4: labelling of figure panels would benefit from being consistent. Either always use "AEL" or "after egg lay (AEL)".

2. Figure labeling:

- a. Several figures lack scale bars in their microscopic images (Figure 1 all panels, Figure 2 all panels, Figure 3G-H, Figure S1 all panels).
- b. It is not clear why Figure 2 shows knock-down of the very same gene multiple times (e.g. 14-3-3e, Dpm53, string). Is each panel derived from a distinct *Drosophila* colony? If not, please remove all redundant panels showing the same gene.
- c. Neither text, nor figure legend nor figure labeling regarding Figure 3 panels G-H' do explain "Ase" staining. Please mention the protein somewhere and explain what it is.
- d. References to figure panels are often not correct, e.g. line 149 in the manuscript text: Figure 3A shows the control and not the brain with shrinkage. Line 150 also references two wrong figure panels (text references Figure 3B-3C, however according to the figure it should be 3G-H'). Please double-check all figure panels and whether they are correctly referenced in the text.
- e. Figure labels in Figure 4A and B are very confusing. To enhance clarity for the reader, I suggest to remove the 48hr AEL label and instead make it very clear that the 2hr, 12hr, 24hr and 48hr timepoints are after IR exposure.

3. Figure presentation:

- a. Adding a schematic showing the hierarchy of events and involved genes/proteins of the DNA repair pathway could enhance clarity for the not so experienced readers and help to make the author's points
- b. Figure 1: it is not clear in the figure and the text why Cul-2 is chosen for a representative gene causing nuclear Pros upon knock-down in NBs. Please indicate the reason or chose a different gene or one gene for each GO Term that was identified.
- c. Figure 3 C-D: here the reader should not expect to see nuclear Pros accumulation. However, the images show a high number of nuclear Pros positive NBs.
- d. Figure 3F and F' do not show the nucleus highlighted by a box in E'. When I compare the boxed nucleus from E' with the nucleus at the same position shown in E and focus on it's surrounding Pros+ cells, then it becomes clear that the nucleus highlighted by dashed lines in F and F' is not the one that is highlighted in E'. Please show the higher magnification images of the correct nucleus.
- e. Figure 4 E-F': please add an image with all 3 markers shown in one photo. With the current presentation it is not possible to spot PH3+ NBs, as there seem to be more PH3+ cells than NBs. What are the non-NB PH3+ cells?
- f. Figure 5 G: result of the statistics test is cut off. Please make it visible.
- g. It is very confusing to read Figure 6. Please chose different types of arrows for NB nuclei showing nuclear Pros and nuclei that don't. Please also add those arrows to the image showing the double staining (panels A, B, C, D).
- h. Figure 7: photos for mre11IR, mei-41IR and WRNexoIR are missing in the figure. Please add them.
- i. Figures S1, 1, 2, and 3 would benefit from showing a quantification of nuclear Pros+ NBs. Please add those quantifications.

4. Statistics

- a. It is not correct to use Student's t-test on graphs that harbor more than 2 datasets. For Figures 4A and B, please chose the correct statistics test (ANOVA).
- b. It is not correct to use the same control dataset in multiple separate graphs. Thus, please show all data sets from Figure 5 in one single graph and apply the correct statistics test (ANOVA).
- c. Please show statistics result for Figure 6 E when comparing all data sets to each other. Please use ANOVA to assess statistics.
- d. Please show statistics result for Figure 7C when comparing all data sets to each other. Please use ANOVA to assess statistics.

5. Additional experiments:

- a. According to the author's hypothesis of premature NB differentiation induced by IR, one would expect a higher number of postmitotic cells in irradiated larval brains as compared to control conditions shortly after IR exposure. Is this assumption correct? Please provide a quantification of postmitotic cells after 24h and 48h after IR exposure. I further suggest to combine this experiment with an EdU pulse to quantify the fraction of EdU+ postmitotic cells and EdU+ NBs.
- b. What happens to the postmitotic daughter cells of irradiated NBs that exhibit DNA damage following knock-down of DNA repair genes? Are they retained within the brain? Are they cleared? The authors might want to use EdU incorporation prior to IR exposure and then trace the fate of EdU+ cells to answer this question.
- c. Please show a gH2av staining in WRNexoIR brains, WRNexo/mre11IR brains and WRNexo/mei-41IR brains to gain deeper insights into the extent of DNA damage upon knock-down of WRNexo.
- d. Please quantify the number of NBs in Pros knock-down brains following IR to prove causal relationship between IR-induced premature NB differentiation and Pros expression

Minor comments

1. Figure 6: Figure is missing the "Figure 6" label
2. Citation Gogondeau et al 2015 (PMID 26573328) is missing, although this work nicely showed that nuclear Pros induces NB differentiation. Please add the reference.
3. The manuscript requires a careful check for typos, spelling and grammar errors.
4. In several sentences, it occurs that the choice of words is semantically not correct, e.g.
 - a. Line 126 word "tanking" - did you intend to say "using"?
 - b. Line 166 word "tested" - did you intend to say "employed"?
 - c. Line 261 word "repair" - did you intend to say "damage"?
5. Please reduced the amount of used words like "then", "this", "they" to enhance clarity for the reader through more crisp language.
6. In general, the writing style could be more precise, e.g. line 161 mentions "decreased" NB numbers, but it is not clear to what the NB numbers are decreased. Adding such detail will greatly enhance clarity for the reader.
7. Line 221 says "BREAK" at the end of the line. I guess this word needs to be removed.
8. The title of the manuscript is very strong. I am not sure one can claim "dominance". I suggest to use a different word that is less hierarchical.

In light of the high number of inconsistencies, I am tempted to request to first fix all inconsistencies and labelling errors and invite the authors to submit a second first submission.

Reviewer #2 (Comments to the Authors (Required)):

In this work, Xu and colleagues explore the role of DNA damage and the repair pathway during the process of neural stem cell renewal and differentiation. A better understanding of the molecular mechanisms that underlie the behavior of stem cells is essential. The authors use the homeodomain transcription factor Prospero (Pros) as a readout of neuroblasts (NB) premature differentiation as this transcription factor regulates the choice between stem cell self-renewal and differentiation. They found, as previously reported by Wagle and Song, 2020, that IR induces premature differentiation of NBs visualized by the nuclear localization of Pros. Using a RNAi screening they demonstrate that the DNA repair pathway plays an important role regulating the maintenance of NBs.

The results are of general interest and important for the stem cell field. The data is presented in a logic manner, and the figures are of good quality. The conclusions are mostly supported by their data. Although I like the paper in general, I found some sentences difficult to read and follow in its current form. Moreover, I suggest the authors to make some corrections in the text and provide some additional data and clarifications before being considered for publication.

Major comments:

- 1-Please, revise the English language of the text, the grammar and the spelling.
- 2-All the gene names should be initially presented with their full name (e.g. ATM= ataxia telangiectasia mutated).
- 3-In general there is a lack of quantifications and experimental description in the text.
- 4-How many samples were analyzed? What is the p-values for each statistical analysis?
- 5-A graph or carton indicating the DNA damage pathway could help the reader follow the logic of the experiments.
- 6-What is the logic behind the selection of the 300 genes for the initial screening? No mention why the authors selected this group of genes.
- 7-Quantifications should be presented for all the genes analyzed in the paper. Any molecular category? What is Cul-2? Is related to their study?
- 8-In most of the paper the authors use the accumulation of nuclear Pros in the NBs as a sign of premature differentiation. However, in their quantifications shown in the figures they reported as the ratio of nuclear Prospero per lobe. This is not correct, and should be indicated as the % of NBs with nuclear Pros, as this gene is expressed also in many cells in the lobe that are not NBs.
- 9-Contrary to what is written in the text, Fig. 1A looks like there are more NBs than the control, is this real? The label of the genotype is not appropriate. It looks like the *worniu-Gal4* is driven the gene white. Also correct the figure legend.
- 10-Fig. 2. Why there are many genotypes repeated? E.g. p53 knockdown is repeated 3 times. Moreover, as the images are presented, it is difficult to observe the nuclear accumulation of Pros. Please include a zoom in those appropriated cases. From their data, only *Mre11* and *CycE-RNAi* show nuclear Pros. Please include numbers for all the genotypes tested.
- 11-Fig. 3: The quantification presented in the figure indicates the number of pH2AV per NBs. However, it could be more informative to indicate how many NBs have pH2AV staining.
- 12-Fig. 4. No data is represented for the IR treatment at 24hrs AEL. This should be shown as the authors claim that almost all NBs show nuclear Pros at this time point (line 159). Only quantifications for the 48hrs AEL are shown.
- In Fig. 4 the reduction of NBs at 48 hrs after IR is minimal when compared to the control no irradiated. However, they compared the n^o of NBs after IR between 24 and 48 hrs, where they found that there is an increase at 24 hrs and then the numbers drop to levels comparable to the control. I think, a more adequate comparison is between the control vs the IR animals at the same time points. These results should be better presented and explained in the text, with the correct comparisons.
- In Fig. S1 the authors must provide a quantification.
- The authors claimed that the drop in NBs number could be due to a mitotic exit and therefore count the number of PH3 positive cells. However, they data presented is for ph3 cells in the entire brain. Is not more accurate to count PH3 cells of NBs to correlate the cell cycle exit of these cells after IR?
- 13-Table 2. Please explain in more detail the data presented and include representative images.
- 14-Fig. 5. Quantifications should be presented relative to the number of NBs and not to the ratio of nuclear Prospero per lobe. Images for *Lig4* and other components of the NHEJ should be presented.
- 15-Fig. 6. The lack of effect after *tefu/ATM* knockdown could be due to the induction of apoptosis in this condition as has been previously reported?
- In the figure they show that the double knockdown of *mei21/ATR* and *tefu/ATM* have a stronger phenotype that the single knockdowns. However, there is no mention in the text about this result.
- Indicate what n stands for and number of brains. Also, indicate that the control is IR also.
- 16-Fig. 7. What is the function of the *WRNexo* in the DNA damage response? Why the authors decide to look at it? Please, provide the logic behind this experiment and discuss it in the text.
- 17-Line 261-264. This sentence could be rewritten to provide a clearer message

Minor comments:

- 18-Revise references to the figures in the text as many do not correspond. E.g. Fig. 3.
- 19-Is more correct to use orthologue instead of homologue when indicating genes from other species.
- 19-How Pros regulates NBs cell cycle exit? Maybe a couple of sentences in the discussion could provide a general picture of NBs response to IR.

Re: Life Science Alliance manuscript #LSA-2022-01802-T

HR repair pathway plays a crucial role in maintaining neural stem cell fate under irradiation stress

Author's Response

Dear Editor

Thank you very much for sending us the supportive and helpful reviewer's comments for our manuscript entitled "HR repair pathway plays a crucial role in maintaining neural stem cell fate under irradiation stress". We have carefully revised our manuscript according to their recommendations. Please see our point-by-point response letter to the comments and suggestions provided by the reviewers. The reviewer's comments are provided in bold text, and our responses in normal text. The revised contents have been marked in blue in the revised manuscript. We sincerely hope our revised manuscript is now satisfactory and acceptable.

Sincerely yours

Xi et al.,

Response to Reviewer

Reviewer's comments to author

Reviewer #1 (Comments to the Authors (Required)):

In their manuscript entitled "MRN complex and the HR repair pathway dominate in maintaining the fate of neural stem cells", Xu et al. describe the activation of DNA repair pathways in neuroblasts (NBs) of the developing *Drosophila* brain as a consequence of exposure to X-ray irradiation (IR).

The authors show a correlation between IR-induced nuclear accumulation of Prospero and loss NB numbers. Thus, Xu et al. speculate that IR induces premature differentiation of NBs. They suggest that premature terminal differentiation represents a preventive cellular mechanism that protects the organism from retainment of damaged or mutant stem cells which might otherwise develop into tumor cells.

Knock-down experiments using RNAi to reduce the expression of several DNA damage repair genes result in increased numbers of NBs showing nuclear accumulation of Prospero and an overall decrease of NB numbers, suggesting that NBs undergo terminal differentiation and are thus lost from the stem cell

pool. The authors conclude that the DNA repair pathway plays an important role in the clearance of IR-induced DNA damage and subsequently in the prevention of premature NB differentiation. Using further RNAi screens, Xu et al. uncover that DNA repair occurs in an ATR- and WRNexo-dependent manner.

Major comments:

The manuscript shows a lot of inconsistencies, labelling errors and confusing figure presentation. Semantically, the text often infers causal relationship while at best the authors show correlations. I strongly invite the authors to adjust their writing style and claims to the content of their data. Given several other published reports showing a role of IR in inducing premature NB differentiation I suggest further experiments to lift the current manuscript to a higher level of novelty.

Response:

Thank you very much for your insightful comments. We have carefully revised our manuscript according to your valuable suggestions aiming to improve the unclear description of our text and the confusing presentations of some of our figures please see our point-to-point response beyond each comment.

1. Inconsistencies:

a. The main text describing results of figure 2 (line 138) claims that "each of these" genes result in nuclear Pros. However, the images and also the legend of Figure 2 show that only two of the tested genes result in nuclear Pros. Please correct the text in line 138 accordingly.

Response:

We are sorry for the unclear description and confusing reference to our previous Figure 2 in the text. Please allow us to explain further. In our study, we performed three sets of RNAi screening. In the first screening of 300 genes, the knockdown of many presented with the nuclear Pros phenotype in NBs (Table 1). Some cases resulted in a high percentage of NBs with nuclear Pros phenotype, which included those of DNA damage repair genes. We then performed the second RNAi screening, combined with X-ray irradiation (IR) treatment, to investigate whether the nuclear Pros phenotype of NBs was related to DNA damage repair pathway genes that activate in response to IR stress. Results showed that under RNAi condition alone, some cases of DNA damage repair related gene knockdown showed nuclear Pros, whilst others did not, when combined with IR treatment, all of these gene knockdown had the nuclear Pros phenotype (Table 2, column 3). In our third screening, we then focused on the NHEJ and HR repair pathways. For clarification, we have now replaced the reference to Figure 2 by the reference to Table 2 in the text, and moved Figure 2 into supplementary figures shown as Figure S1. We have revised the manuscript, on Page 7, lines 142-149.

b. In many parts of the manuscript, the authors state that tefu knock-down does not result in significantly increased nuclear Pros and premature NB differentiation (e.g. Figure 2, Table 2). However, in Figure 6 they show a significant increase of nuclear Pros in tefu knock-down brains. Please elaborate on these contrary data and correct the data display or interpretation.

Response:

tefu knockdown alone does not result in significantly increased nuclear Pros and premature NB differentiation (Table 2, Figure S1 of the revised manuscript). In Figure 6 (shown as Figure 5 in the revised manuscript), we show a significant increase of nuclear Pros of NBs in tefu knockdown combined with IR treatment. Thus, IR treatment was the reason for this discrepancy. We are sorry the original panel labels were not clear, these have been corrected in our revised manuscript.

c. Figure 3 panels G-H' do not have a DAPI signal in the demarcated nuclei. Please elaborate why there is no DAPI and how you determined that the respective demarcated area is a nucleus if there is no DAPI signal?

Response:

The Asense (Ase) is a transcription factor and a marker for NBs. Therefore, we did not use DAPI to indicate the demarcated nuclei, instead, using the Ase signal in this figure to determine the nucleus localized γ H2Av signals. It is shown in Figure 2 of the revised manuscript. We have revised the text on Page 8, lines 159-162.

d. Figure 4, panels A-B: the control brain shown in A contains more nuclear Pros positive NBs than the irradiated brain shown in B. How does this fit with your data and the main message of your manuscript? Please replace the image shown in A with a more representative image. However, if A is representative you might want to rethink the main message of the paper. Similarly, images shown in Figure 3 panels C and E show similar amount of nuclear Pros positive NBs to me.

Response:

Figure 4, panel A (shown as Figure 3A in the revised manuscript) is the count of the total number of NBs in the brains after IR treatment, while panel B is the count of the number of NBs with the nuclear Pros phenotype after IR treatment. Figure 3 panel C shows the absence nuclear Pros in NBs of the control brains without IR treatment (arrow heads), and panel E showed that there are some NBs with nuclear Pros (arrows). We have revised the figure panels as shown in Figure 3, and the legend.

e. Figure 4: labelling of figure panels would benefit from being consistent. Either always use "AEL" or "after egg lay (AEL)".

Response:

Thank you for pointing this out. We apologize for the inconsistency. We have re-labelled the figure panels and unified the use of AEL shown in Figure 3 in the revised manuscript.

2. Figure labeling:

a. Several figures lack scale bars in their microscopic images (Figure 1 all panels, Figure 2 all panels, Figure 3G-H, Figure S1 all panels).

Response:

We thank the reviewer for pointing this out. We have added scale bars in the microscopic images of all panels in Figure 1, Figure S1, Figure 2G-2H, and Figure S2 in the revised manuscript.

b. It is not clear why Figure 2 shows knock-down of the very same gene multiple times (e.g. 14-3-3e, Dpm53, string). Is each panel derived from a distinct *Drosophila* colony? If not, please remove all redundant panels showing the same gene.

Response:

We are sorry for not having labeled the panels of Figure 2 appropriately (Figure S1 of the revised manuscript). You are correct that each panel is derived from a distinct *Drosophila* colony and resulting from different RNAi-lines. Each RNAi-line presents a different target site of the gene. We have collected as many RNAi lines as we can for each gene to exclude the possible off-target effect of RNAi and to avoid background noise in our experiments. We have added the description and clarification of this into the legend of Figure S1 and revised the main text accordingly, on Page 7, lines 142-149.

c. Neither text, nor figure legend nor figure labeling regarding Figure 3 panels G-H' do explain "Ase" staining. Please mention the protein somewhere and explain what it is.

Response:

Ase is a marker of *Drosophila* type I NBs. In the third instar larval brain, the number of type I NBs is about 100 per hemisphere, which are Ase positive, while the number of type II NBs is only 8 per hemisphere, which are Ase negative. In addition, it is notable that Dpn is another nuclear-localized marker for both type I and type II NBs. In our study, we used Dpn as a NB marker for most of the figures of our experiments, and only used Ase as a NB marker for co-staining with γ H2AV in Figure 3G-3H' (Figure 2G-2H' of the revised manuscript) and with PH3 in Figure 4E-4F' (Figure 3G-3H' of the revised manuscript) to avoid the conflict of the antibodies immunized from the same species. We have added clear descriptions on the use of Ase in the revised manuscript, on Page 8, Lines 159-162.

d. References to figure panels are often not correct, e.g. line 149 in the manuscript text: Figure 3A shows the control and not the brain with shrinkage. Line 150 also references two wrong figure panels (text references Figure 3B-3C, however according to the figure it should be 3G-H'). Please double-check all figure panels and whether they are correctly referenced in the text.

Response:

We are sorry for these errors. We have now corrected the references to figure panels of all Figures in the revised manuscript.

e. Figure labels in Figure 4A and B are very confusing. To enhance clarity for the reader, I suggest to remove the 48hr AEL label and instead make it very clear that the 2hr, 12hr, 24hr and 48hr timepoints are after IR exposure.

Response:

We are sorry for the confusing of the labels in Figure 4A-4B (Figure 3A-3B in the revised manuscript). We have removed the 48hr AEL label and made it clear that the 2hr, 12hr, 24hr and 48hr time points are after IR treatment at the stage of 48hr AEL.

3. Figure presentation:

a. Adding a schematic showing the hierarchy of events and involved genes/proteins of the DNA repair pathway could enhance clarity for the not so experienced readers and help to make the author's points.

Response:

We thank the reviewer for this suggestion. We have added a schematic showing the hierarchy of events and involved genes/proteins of the DNA repair pathway, as shown in the Figure 5G of the revised manuscript.

b. Figure 1: it is not clear in the figure and the text why Cul-2 is chosen for a representative gene causing nuclear Pros upon knock-down in NBs. Please indicate the reason or chose a different gene or one gene for each GO Term that was identified.

Response:

Cullin-2 (Cul2) belongs to ubiquitin ligase genes. The ubiquitin regulatory pathway was also one of key pathways searched in our first-round screening for the phenotype of nuclear Pros in NBs. In our first screening, there were many genes knockdown that presented the phenotype of pros nuclear localization in NBs (Table 1), among these, Cul2-RNAi presented the most marked phenotype of Pros nuclear localization. We therefore selected the Cul-2 image, and also added an image of BLM (Figure 1C-1C'' in the revised manuscript) to represent the phenotype of nuclear Pros in NBs. We also noticed that many other genes knockdown had the NB phenotype in larval brains, including those of the DNA damage repair genes. These became of focal

interest. We then performed a second screening focused on DNA damage repair genes (Table 2). In a third screening, we focused on the NHEJ and HR repair pathways. We have revised the explanation of our manuscript on Page 7, Lines 136-138.

c. Figure 3 C-D: here the reader should not expect to see nuclear Pros accumulation. However, the images show a high number of nuclear Pros positive NBs.

Response:

As Pros is also expressed in differentiated neurons, which present as the much smaller-sized cells with Pros positive signals (Figure 2C-2D in the revised manuscript). The microscopic images of larval brains in this Figure were taken as Z-stack superimposed images in order to better show the total number of NBs with Dpn-labeled positive signal, and the number of NBs with both Dpn-positive and nuclear Pros signals in each hemisphere under IR condition (Figure 2E-2F'). There was no NB with both Dpn-positive and nuclear Pros signal in each hemisphere of wild type larval brains (Figure 2C-2D).

d. Figure 3F and F' do not show the nucleus highlighted by a box in E'. When I compare the boxed nucleus from E' with the nucleus at the same position shown in E and focus on it's surrounding Pros+ cells, then it becomes clear that the nucleus highlighted by dashed lines in F and F' is not the one that is highlighted in E'. Please show the higher magnification images of the correct nucleus.

Response:

In Figure 3E-3E (Figure 2E-2E in the revised manuscript), the brain images were taken as Z-stack superimposed images where small Pros-positive cells in other layers are superimposed upon this location. Conversely, figure F is a photogram taken as a single layer to better view the NB Pros phenotype.

e. Figure 4 E-F': please add an image with all 3 markers shown in one photo. With the current presentation it is not possible to spot PH3+ NBs, as there seem to be more PH3+ cells than NBs. What are the non-NB PH3+ cells?

Response:

We thank the reviewer for this recommendation. We have added an image with all 3 markers shown in one photo in Figure 4E''-4F''(Figure 3E''-4F'' in the revised manuscript) of the revised manuscript. In Figure 3E''-4F'', it is shown that there are more PH3+ cells than NBs in the entire brain. This is because that during the first division of NBs, along with NB self-renewal, smaller-sized daughter Ganglion Mother Cells (GMCs) are formed, and also undergo mitosis division. GMCs can also be labeled with the PH+ signal. The non-NB PH3+ cells are therefore considered to be GMCs.

f. Figure 5 G: result of the statistics test is cut off. Please make it visible.

Response:

We have revised the previous Figure 5G and showed it as Figure 5F in the revised manuscript.

g. It is very confusing to read Figure 6. Please chose different types of arrows for NB nuclei showing nuclear Pros and nuclei that don't. Please also add those arrows to the image showing the double staining (panels A, B, C, D).

Response:

We thank the reviewer for this suggestion. We have now used two different types of arrows to mark the NBs with nuclear Pros (yellow arrow) and NBs without nuclear Pros (white arrow), and added the yellow and white arrows to the image showing the double staining in the panels A, B, C, D. We have revised this Figure as shown in Figure 5 and its legend correspondingly in the revised manuscript.

h. Figure 7: photos for mre11IR, mei-41IR and WRNexoIR are missing in the figure. Please add them.

Response:

We have added the photos for the separate knockdown of Mre11, mei-41 and WRNexo under IR condition as shown in Figure 6 A-6C in the revised manuscript.

i. Figures S1, 1, 2, and 3 would benefit from showing a quantification of nuclear Pros+ NBs. Please add those quantifications.

Response: We have showed a quantification of nuclear Pros+ NBs of the figures in the revised manuscript.

4. Statistics

a. It is not correct to use Student's t-test on graphs that harbor more than 2 datasets. For Figures 4A and B, please chose the correct statistics test (ANOVA).

Response:

We are sorry for the confusing presentation of Figure 4 (Figure 3 in the revised manuscript). We have counted NBs with the nuclear Pros phenotype from all experimental groups and controls (each with 10 brains), and compared the percentage of NBs with the nuclear Pros phenotype between successive two-time points. Each analysis was performed from each one dataset. Therefore, we used Student's t-test to compare the phenotype between two-time points, and we also performed an ANOVA to analysis the variation of the phenotype among all time-points. We have revised the presentations of figure panels and the legend in Figure 3 in the revised manuscript to clarify this.

b. It is not correct to use the same control dataset in multiple separate graphs. Thus, please show all data sets from Figure 5 in one single graph and apply the correct statistics test (ANOVA).

Response:

We thank the reviewer for this suggestion. We have now shown all data sets in one single graph (shown as Figure 4 in the revised manuscript). As we conducted fly crossing experiments within the same time as the control group, we therefore used the one control dataset. We compared the percentage of NBs with the nuclear Pros phenotype between each separated gene knockdown and the wild type (each with 10 brains). Each analysis was performed from a different single dataset. Therefore, we used Student's t-test for each two-groups analyzing, and also performed an ANOVA to analyze the variation of the NB phenotype among all different genotypes. We have revised the presentations of figure panels and the legend in Figure 4 in the revised manuscript to correspond to this.

c. Please show statistics result for Figure 6 E when comparing all data sets to each other. Please use ANOVA to assess statistics.

Response:

Thank you. We have added statistical results for Figure 6E (Figure 5 E in the revised manuscript). We have compared the percentage of NBs with the nuclear Pros phenotype between the gene knockdown and the wild type (each with 10 brains). Each analysis was performed from one set of data. Therefore, we used Student's t-test for each, and also performed an ANOVA to analyze the variation of the NB phenotypes among all different genotypes. We have revised the legend in Figure 5E of the revised manuscript to correspond to this.

d. Please show statistics result for Figure 7C when comparing all data sets to each other. Please use ANOVA to assess statistics.

Response:

Thank you. We have now shown statistics result for Figure 7C (Figure 6F in the revised manuscript), comparing all data sets to each other. We also compared the percentage of NBs with the nuclear Pros phenotype between the gene knockdown and the wild type (each with 10 brains). Each analysis was performed from one set of data. Therefore, we used Student's t-test for each, and also performed an ANOVA to analyze the variation of the NB phenotype among all different genotypes. We have revised the legend in Figure 6F of the revised manuscript, to correspond to this.

5. Additional experiments:

a. According to the author's hypothesis of premature NB differentiation induced by IR, one would expect a higher number of postmitotic cells in irradiated larval

brains as compared to control conditions shortly after IR exposure. Is this assumption correct? Please provide a quantification of postmitotic cells after 24h and 48h after IR exposure. I further suggest to combine this experiment with an EdU pulse to quantify the fraction of EdU+ postmitotic cells and EdU+ NBs.

Response:

Under normal conditions, the NBs undergo asymmetric divisions to self-renew and produce smaller-sized daughter cells, namely Ganglion Mother Cells (GMCs). The GMCs then continue differentiation to produce two glia or two neurons. Therefore, it would be expected that all GMCs, glia and neurons will be labeled with EdU in these conditions. However, under IR condition, as some of NBs terminated their cell fates earlier by undergoing premature differentiation, the total number of NBs, GMCs, glia and neurons should be reduced and the EdU labeled postmitotic cells would show corresponding reduction after IR exposure. We have conducted these experiments with EdU pulse after 24h and 48h after IR exposure. As shown in Figure S3, the EdU signals do show significant decreases, compared to the controls. We have added these data in the Results on Page 10, lines 200-204.

b. What happens to the postmitotic daughter cells of irradiated NBs that exhibit DNA damage following knock-down of DNA repair genes? Are they retained within the brain? Are they cleared? The authors might want to use EdU incorporation prior to IR exposure and then trace the fate of EdU+ cells to answer this question.

Response:

In our observation, the postmitotic daughter cells of irradiated NBs that exhibit DNA damage following knock-down of DNA repair genes may be retained rather than cleared, as we did not observe any accumulation of casepase3 signals in the brain under IR treatment (Figure 3A).

c. Please show a γ H2av staining in WRNexoIR brains, WRNexo/mre11IR brains and WRNexo/mei-41IR brains to gain deeper insights into the extent of DNA damage upon knock-down of WRNexo.

Response:

Two WRNexo mutants have already been reported previously (Bolterstein E et al., 2014; Shen J C et al., 2000), in which WRNexo^{e04496} causes a severe reduction in WRNexo expression. WRNexo^{e04496} flies exhibit high sensitivity to the topoisomerase I inhibitor as well as hyper-recombination. WRNexo^{D229V} contains a point mutation that ablates exonuclease activity at physiological temperatures (Shen J C et al., 2000). Like WRNexo^{e04496}, WRNexo^{D229V} mutants also display hyper-recombination. Bolterstein E et al., 2014 has reported that γ H2VA signals were increased in embryos with WRNexo^e mutation. We hypothesize that WRNexo RNAi knockdown in NB also causes hyper-recombination in *Drosophila's* NB. In this case, the γ H2AV signals of

WRNexoIR brains, WRNexo/mre11IR brains and WRNexo/mei-41IR brains would be either reduced or unchanged due to the hyper-recombination activity rescuing the phenotype caused by X-ray with other HR component knockdown. We therefore respectfully suggest that further staining is not a strong focal requirement for the current study. We have revised the text relating to the role of WRNexo in the revised manuscript on on Page 12-13, lines 254-262, Page 13, lines 269-271, and the Discussion of the manuscript, on Page 14-15, lines 293-360.

[1] Bolterstein E, Rivero R, Marquez M, Mcvey M. The *Drosophila* Werner exonuclease participates in an exonuclease-independent response to replication stress[J]. *Genetics*, 2014, 197(2): 643-52.

[2] Shen J C, Loeb L A. The Werner syndrome gene: the molecular basis of RecQ helicase-deficiency diseases[J]. *Trends Genet*, 2000, 16(5): 213-20.

d. Please quantify the number of NBs in Pros knock-down brains following IR to prove causal relationship between IR-induced premature NB differentiation and Pros expression

Response:

Pros is a NB cell fate determinant. Knockdown of Pros results in severe defects in NB division and differentiation, presenting with tumor-like over-proliferation of cells which are not identical to NBs in *Drosophila* brains (ref). IR-induced premature NBs and these proliferative cells in Pros knock-down brains are therefore not directly comparable.

Reference:

Xu, X., X. Wan and X. Wei, 2017 PROX1 promotes human glioblastoma cell proliferation and invasion via activation of the nuclear factor-kappaB signaling pathway. *Mol Med Rep* 15: 963-968.

Minor comments

1. Figure 6: Figure is missing the "Figure 6" label

Response:

We have added the label as shown in Figure 5 in the revised manuscript.

2. Citation Gogendeau et al 2015 (PMID 26573328) is missing, although this work nicely showed that nuclear Pros induces NB differentiation. Please add the reference.

Response:

We thank the reviewer for pointing out this oversight. Gogendeau et al 2015 (PMID 26573328) has now been cited and added into the References of our revised manuscript.

3. The manuscript requires a careful check for typos, spelling and grammar errors.

Response:

We have carefully checked for typos, spelling and grammar errors in our revised manuscript.

4. In several sentences, it occurs that the choice of words is semantically not correct, e.g.

a. Line 126 word "tanking" - did you intend to say "using"?

Response:

We have changed the word "tanking" into "using" in the revised manuscript.

b. Line 166 word " tested " - did you intend to say " employed " ?

Response:

We have changed the word " tested " into " employed " in the revised manuscript on Page 8, line 167.

c. Line 261 word " repair " - did you intend to say " damage " ?

Response:

Yes, We have changed the word " repair " into " damage " in the revised manuscript.

5. Please reduced the amount of used words like "then", "this", "they" to enhance clarity for the reader through more crisp language.

Response:

We have revised the manuscript with reducing the use of "then", "this", "they". We have checked carefully throughout the text.

6. In general, the writing style could be more precise, e.g. line 161 mentions "decreased" NB numbers, but it is not clear to what the NB numbers are decreased. Adding such detail will greatly enhance clarity for the reader.

Response:

We have modified the writing style and added the detail information as following: With a dose of 30Gy X-Ray irradiation (IR) exposure for 11 mins at the stage of 48 hours after egg lay (AEL), we dissected the larval brains at 2hr, 12hr, 24 hr and 48hr points after IR treatment, and examined the phenotype of NBs with nuclear Pros in both experimental groups and control groups. The results showed that at 24hr-point, the brain presented with most NBs showing nuclear Pros. At the 48 hr-point, both the total number of NBs (Figure 3A, right) and the percentage NBs with nuclear Pros had decreased (Figure 3B, right), comparing to that at 24 hr-point. The control groups do not show these effects in either the NBs number (Figure 3A, left) or the Pros

phenotype (Figure 3B, left). We have revised the corresponding text on Page 8-9, Lines 168-178.

7. Line 221 says "BREAK" at the end of the line. I guess this word needs to be removed.

Response:

We have deleted the word "BREAK" in the revised manuscript.

8. The title of the manuscript is very strong. I am not sure one can claim "dominance". I suggest to use a different word that is less hierarchical.

Response:

As our study revealed that the MRN complex and HR repair pathway, rather than non-homologous end-joining (NHEJ) pathway, play a dominant role in the maintenance of NBs under IR stress. We have revised the title of the manuscript, as "HR repair pathway plays a crucial role in maintaining neural stem cell fate under irradiation stress". We hope this title would now be considered appropriate and acceptable.

In light of the high number of inconsistencies, I am tempted to request to first fix all inconsistencies and labelling errors and invite the authors to submit a second first submission.

Response:

We have taken on board this comment and have endeavored to clarify the points that have been raised, some requiring error correction or clarifications to our manuscript, (which we have incorporated and corrected). We truly appreciate reviewer 1 for pointing these out and feel our manuscript has been improved in response to his/her comments. However, some other points, linked to claims of inconsistency, may actually represent misunderstandings or misinterpretation. We hope the above responses and our careful revision of the manuscript have clarified such issues. We appreciate the effort and suggestions of both reviewers and hope the revised manuscript has been satisfactorily improved.

Reviewer #2 (Comments to the Authors (Required)):

In this work, Xu and colleagues explore the role of DNA damage and the repair pathway during the process of neural stem cell renewal and differentiation. A better understanding of the molecular mechanisms that underlie the behavior of stem cells is essential. The authors use the homeodomain transcription factor Prospero (Pros) as a readout of neuroblasts (NB) premature differentiation as this transcription factor regulates the choice between stem cell self-renewal and differentiation. They found, as previously reported by Wagle and Song, 2020, that IR induces premature differentiation of NBs visualized by the nuclear localization of Pros. Using a RNAi screening they demonstrate that the DNA repair pathway plays an important role regulating the maintenance of NBs. The results are of general interest and important for the stem cell field. The data is presented in a logic manner, and the figures are of good quality. The conclusions are mostly supported by their data. Although I like the paper in general, I found some sentences difficult to read and follow in its current form. Moreover, I suggest the authors to make some corrections in the text and provide some additional data and clarifications before being considered for publication.

Response:

We thank the reviewer very much for their comments and suggestions. We have made corrections throughout our manuscript, and provided additional data and figures that have helped us to improve our manuscript. The following are our point-to-point responses.

Major comments:

1-Please, revise the English language of the text, the grammar and the spelling.

Response:

We have carefully revised the English language of the text, the grammar and the spelling, in the revised manuscript.

2-All the gene names should be initially presented with their full name (e.g. ATM= ataxia telangiectasia mutated).

Response: We have revised all the gene names with their full name initially presented in the revised manuscript.

3-In general there is a lack of quantifications and experimental description in the text.

Response:

In accordance with this and reviewer 1's previous comments, we have added the quantifications and experimental description in the revised manuscript.

4-How many samples were analyzed? What is the p-values for each statistical analysis?

Response:

About 10 brains were analyzed for each genotyped flies in each experiment and control group. P-values are now presented in all quantification graphs in Figure 3, Figure 4, Figure 5, and Figure 6, in the revised manuscript.

5-A graph or carton indicating the DNA damage pathway could help the reader follow the logic of the experiments.

Response:

We have added a graph showing the hierarchy of events and involved genes/proteins of the DNA repair pathway, as shown as Figure 5G and the corresponding figure legend.

6-What is the logic behind the selection of the 300 genes for the initial screening? No mention why the authors selected this group of genes. Transcription, DNA Pros

Response:

Our initial determination for the study was to screen the genes related to the premature nuclear Pros in NBs. We selected the 300 genes for the initial screening partially based on the article 'Genome-wide analysis of self-renewal in Drosophila neural stem cells by transgenic RNAi' (Neumüller RA et al, Cell Stem Cell, 2011), and our previous work (Wu et al., Aging Cell,2019) . We focused on the genes mainly functioning in the process of transcriptional regulation, cytoplasmic transportation, and genomic instability during the self-renewal and differentiation of NBs. We have revised manuscript on Page 7, lines 142-149.

7-Quantifications should be presented for all the genes analyzed in the paper. Any molecular category? What is Cul-2? Is related to their study?

Response:

Quantifications have been presented for all the genes analyzed in the revised manuscript.

Cullin-2 (Cul2) belongs to ubiquitin ligase genes. The ubiquitin regulatory pathway was also one of key pathways searched in our first-round screening for the phenotype of nuclear Pros in NBs. In our first screening, there were many genes knockdown that presented the phenotype of pros nuclear localization in NBs (Table 1), among these, Cul2-RNAi presented the most marked phenotype of Pros nuclear localization. We therefore selected the Cul 2 image (Figure 1B-1B'), and also added an image of Blm (Figure 1C-1C'' in the revised manuscript) to represent the phenotype of nuclear Pros in NBs. We also noticed that many other genes knockdown had the NB phenotype in

larval brains, including those of the DNA damage repair genes. These became of our focal interest. We then performed a second screening focused on DNA damage repair genes (Table 2). In a third screening, we focused on the NHEJ and HR repair pathways. We have revised the explanation of our manuscript on Page 7, Lines 136-138.

8-In most of the paper the authors use the accumulation of nuclear Pros in the NBs as a sign of premature differentiation. However, in their quantifications shown in the figures they reported as the ratio of nuclear Prospero per lobe. This is not correct, and should be indicated as the % of NBs with nuclear Pros, as this gene is expressed also in many cells in the lobe that are not NBs.

Response:

We thank the reviewer for this suggestion. We have now shown the quantifications to be indicated as the % of NBs with nuclear Pros in the figures (Figure 3, Figure 4, Figure 5, and Figure 6) of the revised manuscript.

9-Contrary to what is written in the text, Fig. 1A looks like there are more NBs than the control, is this real? The label of the genotype is not appropriate. It looks like the *worniu-Gal4* is driven the gene white. Also correct the figure legend.

Response:

Thank you for pointing this out. We have revised the panels of Figure 1 and replaced the Figure 1A by a Z-stack photo (previously we presented a single layer image), which is matched with the images style of other panels, and corrected the figure legend.

10-Fig. 2. Why there are many genotypes repeated? E.g. p53 knockdown is repeated 3 times. Moreover, as the images are presented, it is difficult to observe the nuclear accumulation of Pros. Please include a zoom in those appropriated cases. From their data, only *Mre11* and *CycE-RNAi* show nuclear Pros. Please include numbers for all the genotypes tested.

Response:

In Figure 2, each panel is derived from a distinct *Drosophila* strain. They are all different RNAi-lines. Each panel is derived from a distinct *Drosophila* colony, all resulting from different RNAi-lines. Each RNAi-line presents a different target site of the gene. We collected as many RNAi lines as we could for each gene to exclude the possible off-target effect of RNAi and to avoid background noise in our experiments. We have added and clarified the description of this in the legend of Figure S1 and into the main text of the revised manuscript, on Page 7, lines 142-149.

11-Fig. 3: The quantification presented in the figure indicates the number of pH2AV per NBs. However, it could be more informative to indicate how many NBs have pH2AV staining.

Response:

Thank you for this suggestion. We have now added the numbers of NBs with γ H2AV staining in the figure legend, shown as Figure 2 in the revised manuscript.

12-Fig. 4. No data is represented for the IR treatment at 24hrs AEL. This should be shown as the authors claim that almost all NBs show nuclear Pros at this time point (line 159). Only quantifications for the 48hrs AEL are shown.

In Fig. 4 the reduction of NBs at 48 hrs after IR is minimal when compared to the control no irradiated. However, they compared the n^o of NBs after IR between 24 and 48 hrs, where they found that there is an increase at 24 hrs and then the numbers drop to levels comparable to the control. I think, a more adequate comparison is between the control vs the IR animals at the same time points. These results should be better presented and explained in the text, with the correct comparisons.

In Fig. S1 the authors must provide a quantification.

The authors claimed that the drop in NBs number could be due to a mitotic exit and therefore count the number of PH3 positive cells. However, they data presented is for ph3 cells in the entire brain. Is not more accurate to count PH3 cells of NBs to correlate the cell cycle exit of these cells after IR?

Response:

We are sorry for the unclear description (previous line 159). With a dose of 30Gy X-Ray irradiation (IR) exposure for 11 mins at the stage of 48 hours after egg lay (AEL), we dissected the larval brains at 2hr, 12hr, 24 hr and 48hr points after IR treatment, and examined the phenotype of NBs with nuclear Pros. The results showed that at 24hr-point, the brain presented with most NBs showing nuclear Pros. At the 48 hr-point, both the total number of NBs (Figure 3A, right) and the percentage NBs with nuclear Pros had decreased (Figure 3B, right), comparing to that at 24 hr-point. The control groups do not show these effects in either the NBs number (Figure 3A, left) or the Pros phenotype (Figure 3B, left). We have revised the corresponding text on Page 8-9, Lines 168-178.

In Fig. S1 (Fig S 2 in the revised manuscript), quantification has now been added.

In Figure 3E"-3F", there are more PH3+ cells than there are NBs in the entire brain. This is because that during the first division of NBs, along with NB self-renewal, its smaller-sized daughter cells, namely Ganglion Mother Cells (GMCs), had formed and also undergone mitosis division. GMCs could also be labeled using PH+ signaling. The non-NB PH3+ cells are GMCs. By observing the mitotic state of the overall NBs and GMCs, we could determine that the cell cycle phase was abnormal after IR treatment, leading to the early termination of NB division and reduction of NB number. We have revised the corresponding legend of Figure 3E"-3F".

13-Table 2. Please explain in more detail the data presented and include representative images.

Response:

We have added more detail to explain the data presented in Table 2 and the representative images in Figure 2 (as shown in Figure S1) of the revised manuscript, on page 7, lines 142-149, and Page 11, lines 216-222.

14-Fig. 5. Quantifications should be presented relative to the number of NBs and not to the ratio of nuclear Prospero per lobe. Images for Lig4 and other components of the NHEJ should be presented.

Response:

The quantifications have been indicated as the % of NBs with nuclear Pros in the figures (Figure 3, Figure 4, Figure 5, and Figure 6 in the revised manuscript). We have added images of Ku80 and CTIp for the other components of the NHEJ, as shown as Figure S 4A-4C, and revised the text on Page 11, lines 216-218.

15-Fig. 6. The lack of effect after tefu/ATM knockdown could be due to the induction of apoptosis in this condition as has been previously reported? In the figure they show that the double knockdown of mei41/ATR and tefu/ATM have a stronger phenotype than the single knockdowns. However, there is no mention in the text about this result. Indicate what n stands for and number of brains. Also, indicate that the control is IR also.

Response:

The previous study reported that the mutation of *Drosophila* tefu/ATM led to neuron and glial cell death in the adult brain (Petersen AJ et al., 2012, Proc Natl Acad Sci U.S.A). It is possible that tefu/ATM knockdown could induce the apoptosis of NBs. However, as shown in Figure 4, and the observation of our previous study (Wu et al., 2019), the nuclear Pros of NBs phenotype is not related to apoptosis. We have added this in the Discussion of the revised manuscript on Page 14, line 283-287.

In Figure 6 (Figure 5 in the revised manuscript), we showed that the statistical results of knocking down tefu alone are not significantly different from those of knocking down mei-41/ ATR, while there is a significant increase in the phenotype under the double knocking down of mei-41 and tefu, compared to the phenotype of the knocking down either one of them alone. This may be due to a redundant, and also a dominant, effect of mei-41. We have added the discussion corresponding in the revised manuscript on Page 14, Lines 288-292.

We have indicated that n stands for NBs and added the number of brains observed, and the control is also under IR treatment in the revised manuscript.

16-Fig. 7. What is the function of the WRNexo in the DNA damage response? Why the authors decide to look at it? Please, provide the logic behind this experiment and discuss it in the text.

Response:

WRN contains a RecQ C-terminal domain and a helicase and ribonuclease D C-terminal (HRDC) domain, which are largely responsible for DNA and protein binding. The RecQ family of helicases are known as the “guardians of the genome” due to their roles in DNA replication, repair, and maintenance of genomic integrity. Like other RecQ family members, WRN exhibits ATP- dependent DNA helicase activity. In *Drosophila melanogaster*, the WRNexo gene encodes a protein with 35% identity and 59% similarity to the exonuclease domain of human WRN. However, WRNexo does not contain a helicase domain. Purified WRNexo exhibits exonuclease activity on single-strand DNA, double-strand DNA with 5’overhangs, and substrates representing replication bubbles. Two WRNexo mutants have already been reported, in which WRNexo⁰⁴⁴⁹⁶ (Bolterstein E et al., 2014), causes a severe reduction in WRNexo expression, resulted from the presence of a piggyBac {RB} transposable element in the 5’-UTR of WRNexo. WRNexo⁰⁴⁴⁹⁶ flies exhibit high sensitivity to the topoisomerase I inhibitor as well as hyper-recombination (Bolterstein E et al., 2014). A second mutant, WRNexo^{D229V} (Shen J C et al., 2000) contains a point mutation that ablates exonuclease activity at physiological temperatures. Like WRNexo⁰⁴⁴⁹⁶, WRNexo^{D229V} mutants display hyper-recombination. We propose that WRNexo RNAi knockdown in NB caused hyper-recombination activity which rescued the phenotype caused by X-ray with other HR components knockdown. WRNexo may function downstream of the DNA sensory process and transduce signals to activate the nuclear Pros accumulation process, which then prematurely regulates neural stem cells to terminate their stem cell fate rather than continue proliferation. We have added this understanding into the Results on Page 12-13, lines 254-262, Page 13, lines 269-271, and the Discussion of the manuscript, on Page 14-15, lines 293-360.

References:

- [1] Bolterstein E, Rivero R, Marquez M, Mcvey M. The *Drosophila* Werner exonuclease participates in an exonuclease-independent response to replication stress[J]. *Genetics*, 2014, 197(2): 643-52.
- [2] Shen J C, Loeb L A. The Werner syndrome gene: the molecular basis of RecQ helicase-deficiency diseases[J]. *Trends Genet*, 2000, 16(5): 213-20.

17-Line 261-264. This sentence could be rewritten to provide a clearer message

Response:

We thank the reviewer for pointing this out. We have rewritten this sentence in the Discussion on Page 15, lines 317-320.

Minor comments:

18-Revise references to the figures in the text as many do not correspond. E.g. Fig. 3.

Response:

We have revised all references to the figures in the text in the revised manuscript.

19-Is more correct to use orthologue instead of homologue when indicating genes from other species.

Response: We have corrected this and used orthologue instead of homologue when indicating genes from other species.

19-How Pros regulates NBs cell cycle exit? Maybe a couple of sentences in the discussion could provide a general picture of NBs response to IR.

Response: During NB asymmetric divisions, Pros, as a NB cell fate determinant, is localized in the cytoplasm of NB and is segregated exclusively into ganglion mother cells (GMCs) during NB asymmetric divisions. At the point when Pros either prematurely enters the nucleus of NBs at the larval stage, or normally enters it at the pupa stage, NBs then exit the cell cycle and terminate their NB cell fate. No mitotic NBs were detected in the central brain or ventral nerve cord of adult flies (Wu et al., 2019). Under IR stress, DNA damage repair defects therefore caused early entry of Pros into the nucleus of NBs which could lead to premature cell cycle exit and termination of NBs asymmetric division. We have added this content to the Discussion of the revised manuscript, on Page 15, lines 307-314.

April 11, 2023

Re: Life Science Alliance manuscript #LSA-2022-01802-TR

Dr. Yongmei Xi
Women's Hospital, School of Medicine, Zhejiang University
Institute of Genetics, Zhejiang University and Department of Genetics, Zhejiang University School of Medicine
Yuhangtang Road 866, Xihu District
Xihu district
Hangzhou, Zhejiang 310058
China

Dear Dr. Xi,

Thank you for submitting your revised manuscript entitled "HR repair pathway plays a crucial role in maintaining neural stem cell fate under irradiation stress" to Life Science Alliance. The manuscript has been seen by the original reviewers whose comments are appended below. While the reviewers continue to be overall positive about the work in terms of its suitability for Life Science Alliance, some important issues remain.

Our general policy is that papers are considered through only one revision cycle; however, given that the suggested changes are relatively minor, we are open to one additional short round of revision. Please note that I will expect to make a final decision without additional reviewer input upon resubmission.

Please submit the final revision within one month, along with a letter that includes a point by point response to the remaining reviewer comments.

To upload the revised version of your manuscript, please log in to your account: <https://lsa.msubmit.net/cgi-bin/main.plex>
You will be guided to complete the submission of your revised manuscript and to fill in all necessary information.

B. MANUSCRIPT ORGANIZATION AND FORMATTING:

Sincerely,

Reviewer #1 (Comments to the Authors (Required)):

The manuscript by Xu et al has significantly improved through the revision. However, I have noticed a few more inaccuracies.

If those minor points will be fixed, the paper is acceptable for publication from my side:

1. Citation styles: The citation styles within the paper do not match. Please double-check whether all references are correctly updated in the manuscript bibliography, e.g. line 129-130, line 185, line 287, line 289-290, line 299, line 303, line 304, line 316-317.
2. The EdU protocol is missing from the methods. Please describe the details of EdU treatment (timepoint of EdU exposure, time duration of EdU pulse, staining procedure, product information).
3. The description of Figures 2C-F is missing from the main text. Please either describe what the figure shows in the main text or remove panels 2C-F from the figure.
4. Main text description of Figure 2I is wrong. The figure does not show Pros staining, but gH2Av+ foci per NB. Please correct the figure description.
5. Figure S3: The main text does not explain why the WRnexo colony is shown in addition? Please either describe the usage of WRnexo flies in the main text or remove the panels S3D-F and the right part of the graph in S3G from the figure.
6. Figure S4: The main text does not explain why CtIP is shown in the figure panels. Please either provide an explanation in the text or remove panels B and the corresponding bar from the figure.
7. Figure S4: please provide a representative image of the control w1118.
8. Main text lines 219-222: the sentence is lacking a reference to a figure that shows the mentioned results. Please add a figure reference.
9. Please don't use unpaired t-test in any graph that harbors more than 3 bars. For such graphs use ANOVA with multiple comparisons.
10. Figure Legend Figure 2I: please indicate p-values.
11. Typo in graph legend in Figure 3B, Figure 4F, Figure 5E, Figure 6F, Figure S4C: it should be nuclei and not nucle
12. Figure 4: image labels "Pros" and "Dpn Pros" are swapped. Please add the image labels to the correct corresponding stainings.

Reviewer #2 (Comments to the Authors (Required)):

I acknowledge the effort that the authors have done to improve the manuscript. However, there are still some errors and concerns that should be corrected/explained before considering this article for publication.

-In the first revision I suggested to represent the quantifications as % of NB with nuclear Pros instead of ratio of nuclear Prospero per lobe. The authors have changed the Y axis label in the graphs, however the quantifications remain the same. I am confused, should not the numbers be different?

-The authors should indicate also in the text the number of brains used for each experiment.

-Figure 2 legend does not show the statistical analysis used, neither the p value.

-The authors have not corrected the label of the genotype in Fig. 1A. As it reads, it looks like the Gal4 is activating the white gene. Moreover, this mistake have not been changed in the figure legend.

-Please include a zoom in Fig. S1 (previous Fig. 2) to help the reader observe the nuclear accumulation of Pros in the appropriate cases. Also, I think it would be informative to include quantifications for the genotypes and treatments in Fig. S1 and table 2 as suggested. Also include representative images for the genotypes in table 2 after IR.

-Fig. 2. As suggested in the first revision it would be more informative to indicate the number of NB that have pH2AV staining, instead of the number of pH2AV foci per NBs

-Line 161: Please correct the reference to the figure.

-Results section 4. The description of the HR pathway should be introduced before the results obtained after the knockdown of HR repair pathway genes.

Re: Life Science Alliance manuscript #LSA-2022-01802-TR

HR repair pathway plays a crucial role in maintaining neural stem cell fate under irradiation stress

Author's Response

Dear Editor

Thank you very much for yours and reviewers' positive comments and suggestions regarding to our manuscript entitled "HR repair pathway plays a crucial role in maintaining neural stem cell fate under irradiation stress". We have carefully revised our manuscript according to these recommendations and editorial requirements for revisions. Please see our point-by-point response letter. The reviewer's comments are provided in bold text, and our responses in normal text. The revised contents have been marked in red in the revised manuscript. We sincerely hope our revised manuscript is now satisfactory and acceptable.

Sincerely yours

Xi et al.,

Response to Reviewer

Reviewer's comments to author

Reviewer #1 (Comments to the Authors (Required)):

The manuscript by Xu et al has significantly improved through the revision. However, I have noticed a few more inaccuracies.

If those minor points will be fixed, the paper is acceptable for publication from my side:

Response:

Thank you very much for your comments. We have carefully revised our manuscript according to your valuable suggestions.

1. Citation styles: The citation styles within the paper do not match. Please double-check whether all references are correctly updated in the manuscript bibliography, e.g. line 129-130, line 185, line 287, line 289-290, line 299, line 303, line 304, line 316-317.

Response:

We have revised the citation styles within the paper and kept all the reference formats consistent.

2. The EdU protocol is missing from the methods. Please describe the details of EdU treatment (timepoint of EdU exposure, time duration of EdU pulse, staining procedure, product information).

Response:

We have added the Edu protocol in the Methods of the revised manuscript, on Page 19-20, lines 405-416.

3. The description of Figures 2C-F is missing from the main text. Please either describe what the figure shows in the main text or remove panels 2C-F from the figure.

Response:

Thank you very much for your comments. We have added description of Figures 2C-2F in the revised manuscript, on Page 8, lines 155-156.

4. Main text description of Figure 2I is wrong. The figure does not show Pros staining, but gH2Av+ foci per NB. Please correct the figure description.

Response:

We are very sorry for the mistake, and have corrected the description of Figure 2I and other figure descriptions of Figure 2 in the revised main text, on Page 8, lines 157-160.

5. Figure S3: The main text does not explain why the WRnexo colony is shown in addition?

Please either describe the usage of WRnexo flies in the main text or remove the panels S3D-F and the right part of the graph in S3G from the figure.

Response:

Since the knockdown of WRNexo in neuroblasts could result in hyper-recombination. The homologous recombination is mainly occurred in the S and G2 phases. Edu indicates the S phase. Therefore, we examined the WRNexo knockdown colony to look at the cell division during the corresponding period. As shown in Figure S3D-S3F, the EdU signals showed no difference at 24hr-point and 48hr-point after IR exposure, suggesting activated HR pathway postponed premature differentiation. We have added the relevant description in the revised manuscript, on Page 13, line 266-270.

6. Figure S4: The main text does not explain why CtIP is shown in the figure panels.

Please either provide an explanation in the text or remove panels B and the corresponding bar from the figure.

Response:

We have added an explanation about CtIP shown in the figure panels, and revised the text on Page 11, line 215-218.

7. Figure S4: please provide a representative image of the control w1118.

Response:

We have provided a representative image of the control w1118, shown in Figure S4A-S4A'.

8. Main text lines 219-222: the sentence is lacking a reference to a figure that shows the

mentioned results. Please add a figure reference.

We have added a reference to Table 2 and Figure S5 in the revised text, on Page 11, lines 231-234 (previous lines 219-222).

9. Please don't use unpaired t-test in any graph that harbors more than 3 bars. For such graphs use ANOVA with multiple comparisons.

Response:

Thank you very much for your suggestion. We have used ANOVA instead of unpaired t-test in the graphs that harbors more than 3 bars, including the Figure 3A-3B, Figure 4F, Figure 5E, Figure 6F, Figure S2E, Figure S3G, Figure S4C, and revised corresponding figure legends and Methods.

10. Figure Legend Figure 2I: please indicate p-values.

Response:

We have indicated P-values in the Figure Legend of Figure 2I, on Page 27, lines 619-620.

11. Typo in graph legend in Figure 3B, Figure 4F, Figure 5E, Figure 6F, Figure S4C: it should be nuclei and not nucle.

Response:

We have corrected the typo in the graph legends of Figure 3B, Figure 4F, Figure 5E, Figure 6C, Figure S4C, in lines 624-625, 642-643, 655, 662, 702

12. Figure 4: image labels "Pros" and "Dpn Pros" are swapped. Please add the image labels to the correct corresponding staining.

Response:

We have revised the swapped image labels "Pros" and "Dpn Pros" in the Figure 4.

Reviewer #2 (Comments to the Authors (Required)):

I acknowledge the effort that the authors have done to improve the manuscript. However, there are still some errors and concerns that should be corrected/explained before considering this article for publication.

Response:

Thank you very much for your comments. We have carefully revised our manuscript according to your valuable suggestions.

1. In the first revision I suggested to represent the quantifications as % of NB with nuclear Pros instead of ratio of nuclear Prospero per lobe. The authors have changed the Y axis label in the graphs, however the quantifications remain the same. I am confused, should not the numbers be different?

Response:

Thank you very much for your comments. We are so sorry for not having described clearly on items of the quantification, in which we did quantified the numbers of NB with nuclear Pros, not the ratio of nuclear Prospero per lobe. In this case the ratio of the numbers of NB with nuclear Pros to total NBs is the same as the percentage of them. We have revised the manuscript on lines 155-156, 171-172, 239, 251, 265-266.

2. The authors should indicate also in the text the number of brains used for each experiment.

Response:

We have indicated the number of brains used for each experiment, in the text and the

legends of figures and tables, in lines 603, 615, 671-672, 676, 683-684, 689-690,696-697,702-703.

3. Figure 2 legend does not show the statistical analysis used, neither the p value.

Response:

We have added the description of P-values to the figure legend of Figure 2, on lines 619-620.

4.The authors have not corrected the label of the genotype in Fig. 1A. As it reads, it looks like the Gal4 is activating the white gene. Moreover, this mistake have not been changed in the figure legend.

Response:

We have corrected the image label in Figure1, and in the figure legend.

5.Please include a zoom in Fig. S1 (previous Fig. 2) to help the reader observe the nuclear accumulation of Pros in the appropriate cases. Also, I think it would be informative to include quantifications for the genotypes and treatments in Fig. S1 and table 2 as suggested. Also include representative images for the genotypes in table 2 after IR.

Response:

We have included zoomed pictures in Fig.S1 with the nuclear accumulation of Pros. The quantification for the genotypes and treatments in Fig. S1 and table 2 have been indicated in the 4th -5th columns of the table 2. We have included representative images for the genotypes in table 2 after IR, in Figure S5.

6.Fig. 2. As suggested in the first revision it would be more informative to indicate the

number of NB that have pH2AV staining, instead of the number of pH2AV foci per NBs.

Response:

Thank you for your suggestion. We are afraid that due to the staining conflict of the same sourced antibodies, we failed to co-stain the pH2AV and Dpn (NB marker) in the same brain samples. Therefore, we could not count the pH2AV positive NBs, but counted the foci of pH2AV in the whole brain to estimate the increase of DNA damage caused by X-Ray.

7. Line 161: Please correct the reference to the figure.

Response:

We have corrected the reference to the figure in Line 159 (previous 161).

8.Results section 4. The description of the HR pathway should be introduced before the results obtained after the knockdown of HR repair pathway genes.

Response:

Thank you very much for your suggestion. We have revised the text and introduced the description of the HR pathway before the results obtained after the knockdown of HR repair pathway genes, on Page 11, lines 219-231.

April 28, 2023

RE: Life Science Alliance Manuscript #LSA-2022-01802-TRR

Dr. Yongmei Xi
Women's Hospital, School of Medicine, Zhejiang University
Institute of Genetics, Zhejiang University and Department of Genetics, Zhejiang University School of Medicine
Yuhangtang Road 866, Xihu District
Xihu district
Hangzhou, Zhejiang 310058
China

Dear Dr. Xi,

Thank you for submitting your revised manuscript entitled "HR repair pathway plays a crucial role in maintaining neural stem cell fate under irradiation stress". We would be happy to publish your paper in Life Science Alliance pending final revisions necessary to meet our formatting guidelines.

- please add ORCID ID for secondary corresponding author-they should have received instructions on how to do so
- please use the [10 author names, et al.] format in your references (i.e. limit the author names to the first 10)

A. FINAL FILES:

B. MANUSCRIPT ORGANIZATION AND FORMATTING:

**Submission of a paper that does not conform to Life Science Alliance guidelines will delay the acceptance of your

manuscript.**

The license to publish form must be signed before your manuscript can be sent to production. A link to the electronic license to publish form will be sent to the corresponding author only. Please take a moment to check your funder requirements.

Sincerely,

May 4, 2023

RE: Life Science Alliance Manuscript #LSA-2022-01802-TRRR

Dr. Yongmei Xi
Women's Hospital, School of Medicine, Zhejiang University
Institute of Genetics, Zhejiang University and Department of Genetics, Zhejiang University School of Medicine
Yuhangtang Road 866, Xihu District
Xihu district
Hangzhou, Zhejiang 310058
China

Dear Dr. Xi,

Thank you for submitting your Research Article entitled "HR repair pathway plays a crucial role in maintaining neural stem cell fate under irradiation stress". It is a pleasure to let you know that your manuscript is now accepted for publication in Life Science Alliance. Congratulations on this interesting work.

DISTRIBUTION OF MATERIALS:

Again, congratulations on a very nice paper. I hope you found the review process to be constructive and are pleased with how the manuscript was handled editorially. We look forward to future exciting submissions from your lab.

Sincerely,
